# STEALING AND DEFENDING TRANSFORMER-BASED ENCODERS

## ABSTRACT

Self-supervised learning (SSL) has become the predominant approach to training on large amounts of unlabeled data. New real-world APIs offer services to generate high-dimensional representations for given inputs based on SSL encoders with transformer architectures. Recent efforts highlight that it is possible to steal high-quality SSL encoders trained on convolutional neural networks. In this work, we are the first to extend this line of work to stealing and defending transformer-based encoders in both language and vision domains. We show that it is possible to steal transformer-based sentence embedding encoders solely using their returned representations and with 40x fewer queries than the number of the victim's training data points. We further decrease the number of required stealing queries for language encoders by reusing extracted representations on semantically similar sentences and for the vision encoders by leveraging semi-supervised learning. Finally, to defend transformers against stealing attacks, we develop a new scheme to watermark the language encoders, where for the last training iterations, we alternate between optimizing for the standard sentence embedding and a chosen downstream task. For the vision domain, we design a defense technique that combines watermarking with dataset inference. Our method creates a unique encoder signature based on a private data subset that acts as a secret seed during training. By applying dataset inference on the seed, we can then successfully identify stolen transformers.

## 1 INTRODUCTION

The success of self-supervised learning (SSL) motivates the emergence of large-scale services offering API access to encoders which return high-dimensional representations for given inputs. These representations serve to train a diverse set of downstream tasks with a small amount of labeled data. Latest APIs (Clarifai, 2022; Cohere, 2022; OpenAI, 2022) use transformer-based encoders (Devlin et al., 2018; Dosovitskiy et al., 2020) to generate representations. Such encoders have a high number of parameters (*e.g.,* the state-of-the-art RoBERTa-Large language encoder (Liu et al., 2019) has roughly 355M parameters) and are trained on datasets consisting of millions of data points—yielding a highly expensive training procedure (Sharir et al., 2020). Therefore, these encoders are lucrative targets for stealing attacks (Tramèr et al., 2016) where an adversary extracts a victim encoder by submitting queries and using the outputs to train a local stolen copy, often at a fraction of the victim's training cost (Sha et al., 2022; Dziedzic et al., 2022a). The stolen encoder can then be used for inferences without the owner's permission, violating their intellectual property right and causing financial loss.

While stealing and defending supervised models has been heavily studied (Tramèr et al., 2016; Juuti et al., 2019; Orekondy et al., 2020), research on the topic of stealing and defending transformer-based encoders is limited. Despite the immediate practical importance of this problem, to the best of our knowledge, all previous works on model stealing attacks and defenses against SSL encoders are conducted offline in contrived experimental settings (Cong et al., 2022; Sha et al., 2022; Dziedzic et al., 2022a;b), focusing on the vision domain with convolutional neural network (CNN)-based architectures, and do not attack the popular transformer (Vaswani et al., 2017) architecture, which currently holds state-of-the-art results on many vision (Caron et al., 2021) and natural language processing (NLP) tasks (Gao et al., 2021). Since transformers are trained differently than CNNs and represent larger and more complex architectures, stealing them is more challenging.

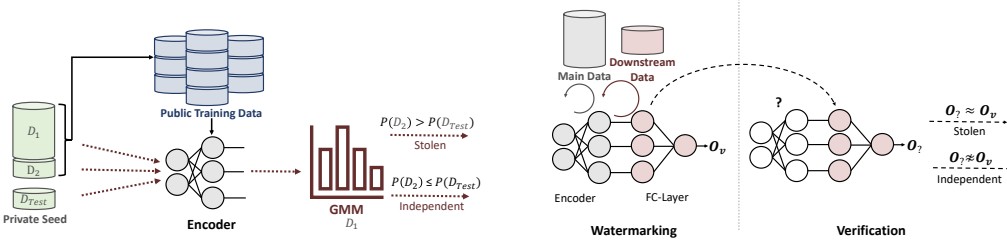

(a) DataSeed Inference (DSI) for Vision Encoders.   (b) Watermark for Language Encoders.

Figure 1: **Defenses against Stealing Transformer-based Encoders. (a)** Our DSI relies on a private dataset, randomly partitioned into three subsets $D_1, D_2, D_{\text{Test}}$. $D_1 \cup D_2$ is injected into the encoder's public training data as a secret seed. For ownership resolution, a Gaussian Mixture Model (GMM) is trained on the representations of $D_1$. DSI marks an encoder as stolen if the average likelihood from the GMM on $D_2$ is significantly higher than the likelihood on $D_{\text{Test}}$. **(b)** To imbed our watermark for sentence embedding encoders, training alternates between the main task and a secretly chosen downstream task during the last iterations. For verification, the fully-connected (FC) layer is attached to a potentially stolen copy and agreement to the victim's output on the downstream task is measured.

We show how stealing attacks (Sha et al., 2022; Dziedzic et al., 2022a) can be successfully applied to extract transformer encoders through their returned representations. In NLP, we are able to successfully steal sentence encoders using only a small number of queries; up to 40x fewer stealing queries than the number of the original training data points. We also show that this number can be further reduced by re-using the representations obtained from the victim encoder for semantically similar sentences of the stealing queries. For vision encoders, we decrease the number of queries against victim encoders by adapting semi-supervision based on MixMatch (Berthelot et al., 2019), which has, so far, only been applied to attacks in the supervised setting (Jagielski et al., 2020).

The successful applicability of encoder stealing to transformer-based architectures in public API settings motivates the urgent need for defenses. We first propose *DataSeed Inference (DSI)*, a combination between watermarking (Uchida et al., 2017; Jia et al., 2021; Adi et al., 2018) and Dataset Inference (DI) (Dziedzic et al., 2022b) as a successful defense for vision encoders. While standard DI operates on the assumption that an encoder is uniquely characterized by its whole training data and exploits that the unique data signature of the victim encoder is transferred to stolen copies, DSI adds a private data subset as a unique seed to the original training data and uses this seed to identify stolen copies. This is a necessary adaption of DI to transformer-based encoders since these are trained on a large amount of *public data*—possibly scrapped from the Internet (Radford et al., 2021), and, thereby not necessarily unique to a single encoder. We note that the signal from the private seed is transferred during stealing. Additionally, our defense does not harm the performance of the defended encoder on downstream tasks, which distinguishes it from watermarks in supervised settings.

Since we find that DSI is not successful in defending language encoders, we propose a new watermarking scheme to protect them from theft. Our watermark relies on alternating between the actual sentence embedding task and a secretly chosen downstream task during the last iterations of training. This transforms the representations so that they preserve their high performance on sentence embedding tasks while increasing their accuracy on the downstream task. To embed the watermark task, we append a fully-connected layer to the encoder. The additional layer acts as our secret key. We verify whether a given encoder is a stolen copy by attaching that layer and checking the agreement to the victim encoder's output for the watermark downstream task. Victim and independent encoders have significantly different outputs whereas victim and stolen copies return similar outputs.

To summarize, we make the following contributions:

- We successfully steal NLP and vision transformer-based encoders in a real-world API setting, assuming access to representations only and using up to 40x fewer queries than the number of samples in the respective encoder's training dataset. Our stolen encoders achieve comparable performance to the respective victims on standard benchmarks.

- We further reduce the number of stealing queries by using semantically similar sentences for language encoders and semi-supervised methods for vision encoders.

- For the vision domain, we propose DSI as a defense which adds a private dataset as a secret seed to the defender's original training data and uses this seed for ownership resolution.
- For NLP sentence embedding transformers, we propose a method to watermark their representations by alternating between the actual sentence embedding and a secretly chosen downstream task during the last iterations of training.

## 2 BACKGROUND AND RELATED WORK

**Model extraction attacks.** The goal of the model extraction attacks is to replicate a functionality of a victim model $f_v$ trained on a dataset $D_P$. An attacker has a black box access to the victim model and uses a stealing dataset $D_s = \{q_i, f_v(q_i)\}_{i=1}^n$, consisting of queries $q_i$ and the corresponding predictions or representations returned by the victim model, to train a stolen model $f_s$. Model extraction attacks have been shown against various types of models including classification (Tramèr et al., 2016; Jagielski et al., 2020) and representation models (Sha et al., 2022; Dziedzic et al., 2022a).

**Language Encoder.** We use SimCSE (Gao et al., 2021) to learn sentence representations since it outperforms other methods and is exposed via public APIs (Clarifai, 2022). The **SimCSE** framework by (Gao et al., 2021) proposes unsupervised and supervised approaches to generate sentence embeddings. It starts from a pre-trained checkpoint of a BERT-based encoder, *e.g.,* RoBERTa, and takes the representation for the classification token ([CLS]) as the sentence embedding. In this work, we rely on their *supervised approach* leveraging pairs of sentences from natural language inference (NLI) datasets within a contrastive learning framework. It uses the *entailment* pairs as positives and *contradictions* as hard negatives.

**Vision Encoder.** In our work, we use **DINO** (Caron et al., 2021) to train vision transformers (ViT) since it returns high-quality representations that achieve SoTA performance on downstream tasks when assessed by training a linear classifier directly on representations. DINO trains student and teacher encoders, both with the same architecture but different parameters, where the teacher is updated with an (exponential moving) average of the student. Different random transformations of the same image are generated and passed through both the student and teacher. The student is provided with smaller crops of the inputs than the teacher, which forces the student to generate representations that restore parts of the initial image. The training objective is to minimize the cross-entropy loss between representations from teacher and student.

**Stealing Encoders.** Thus far, methods for stealing encoders through representations have been shown in the computer vision domain and only for CNNs (Sha et al., 2022; Dziedzic et al., 2022a). Previous work in the NLP domain focuses on classification tasks and performs stealing against fine-tuned models through labels based on a given pre-trained language encoder (Krishna et al., 2020; Zanella-Beguelin et al., 2021; He et al., 2021). Model extraction against NLP APIs are shown by (Xu et al., 2021), specifically for sentiment classification and machine translation tasks. The setup of previous work differs from ours which is concerned with stealing through representations instead of low-dimensional outputs, such as labels. This is motivated by the fact that these representations are exactly what new public APIs expose (Cohere, 2022; Clarifai, 2022). Distillation methods used in the NLP domain (Jiao et al., 2019) which could, in principle, be applied to stealing encoders, usually require white box access to the original model, for example, to the attention layers (Jiao et al., 2019). Therefore, distillation cannot be applied to stealing in public API-access scenarios.

**Defending Encoders.** Dataset inference (DI) (Maini et al., 2021) is a defense against model stealing attacks. It uses the victim's training dataset as a unique signature, leveraging the following observation: for a victim encoder trained on its private data as well as for its stolen copies, the distribution of the representations generated from the victim's training data differs from the distribution of the representations generated on the test data. In contrast, for an independently trained encoder, these two distributions cannot be distinguished, allowing the detection of stolen copies (Dziedzic et al., 2022b). For modeling the distributions, Gaussian Mixture Models (GMMs) are trained on a fraction of the private training data and applied to a disjoint fraction of the training data and the test data. An encoder is identified as a victim or stolen copy if the log-likelihood on private representations is significantly higher than on the test representations. Recently, watermarking (Uchida et al., 2017; Jia et al., 2021; Adi et al., 2018) methods have been proposed for encoders (Dziedzic et al., 2022a; Cong et al., 2022; Wu et al., 2022). The main difference between previous work and our DSI lies in verification. Previous watermarking techniques use downstream tasks to detect a watermark while we resolve

ownership based on the representations directly. For a more detailed overview on watermarking for encoders and a more thorough overview of related work, see Appendix A.

## 3    STEALING TRANSFORMER-BASED ENCODERS

We aim at stealing BERT-based transformers, fine-tuned to return sentence embeddings in the language domain, and general embeddings for images in the vision domain. Our stealing operates in a public API setting where the adversary can query the models through a pre-defined interface to obtain high-dimensional representations for their inputs. Stealing is then performed following previous work (Dziedzic et al., 2022a): (i) The adversary sends $N$ raw or augmented inputs to the victim encoder. These inputs can, in principle, be taken from any data distribution of the target domain, using open-source data. (ii) With the obtained representations, the adversary trains a stolen copy of the victim. The goal of this training is to maximize the similarity of the stolen copy's output and the representations output by the victim. Therefore, the adversary either imitates a self-supervised training using a contrastive loss function, *e.g.,* InfoNCE Chen et al. (2020) or SoftNN Frosst et al. (2019), or directly matches both models' representations via the Mean Squared Error (MSE) loss.

**Stealing Transformer-Based Encoders.** For language, public APIs (Cohere, 2022) expose transformers which are first pre-trained on a large corpus of text data to return per-token representations and then fine-tuned to return high-dimensional embeddings for a given full-text input, *e.g.,* a sentence. In the vision domain, APIs such as (Clarifai, 2022) expose encoders trained from scratch on large amounts of image data to return per-image representations. We find that public APIs (Clarifai, 2022) provide metadata about exposed encoders, which can contain information about datasets used for pre-training as well as the encoder architecture. Thus, we can instantiate our stolen encoders with the victim encoder's architecture. We also follow the API setting and initialize the stolen copies of language encoders with publicly available pre-trained transformers.[1] The stolen copies of vision encoders are initialized with random weights.

**Stealing with Semi-Supervision (Vision).** To reduce the number of stealing queries[2], we apply semi-supervised learning which has been used to improve stealing in the supervised setting (Jagielski et al., 2020). The approach relies on semi-supervision based on MixMatch (Berthelot et al., 2019) and helps to leverage a large pool of unlabeled data while having access to only a small fraction of labeled data (Sohn et al., 2020; Assran et al., 2021). The inspiration for applying MixMatch-based methods to stealing *encoders* comes from DINO (see Section 2), which normalizes representations with a temperature softmax and uses the cross-entropy loss to minimize distances between outputs from student and teacher encoders. Similarly, during stealing, we pass the victim and stolen representations through the softmax layer whose outputs act as our new labels for MixMatch. Note that this method is not applicable to the fine-tuned sentence embedding encoders since these do not rely on data augmentations, which is necessary for MixMatch.

**Re-using Representations (NLP).** For stealing sentence encoders, we reduce the number of stealing queries by re-using representations over semantically similar sentences. This is possible since sentence encoders are required to return similar representations to such semantically similar sentences. Hence, when a stealing dataset holds such similar sentences (*e.g.,* in the form of positive pairs, such as all the datasets used in this work), we only have to query one of these sentences, and assign the same representation to all semantically similar sentences to augment our fine-tuning dataset for the stolen-copy. Our experimental evaluation in Section 5.1 shows the effectiveness of this approach.

## 4    DEFENSES AGAINST STEALING TRANSFORMERS

Given the high training costs of transformer-based encoders, defending them from stealing is an urgent need. Yet, their representations' high dimensionality and complexity make defenses challenging.

---

[1]We use transformers from Hugging Face (`https://huggingface.co/`.

[2]This is relevant in public API settings since costs usually increase linearly with the number of queries and since the number of representations that can be obtained in a given time-unit is often limited.

## 4.1 DataSeed Inference for Vision Transformers

Prior work (Dziedzic et al., 2022b) has successfully applied dataset inference (DI) (Maini et al., 2021) to identify stolen copies of encoders with CNN-architectures. To verify ownership, DI assumes the training data to be *private* in order to act as a unique model signature. However, transformers are usually trained on millions of *public* data points, including noisy and uncurated data scrapes from the Internet (Radford et al., 2021). Thus, large transformers are trained on overlapping datasets which makes DI non-applicable in this setting. To overcome this difficulty and leverage transformers' training signatures for ownership resolution, we propose *DataSeed Inference (DSI)*. This defense combines DI with watermarking (Uchida et al., 2017) by including private data as a secret *seed* into the original training data and training the defended transformer on the combined datasets. The private seed then serves to uniquely identify encoders and their stolen copies. Even if an adversary is aware of our defense, they still require knowledge about the private seed to avoid extracting features contained in it. After stealing an encoder with an arbitrary dataset from the same data domain, we use the private data as well as its corresponding private test data for DSI. We visualize our approach in Figure 1a. An advantage of DSI is that the private subset does not harm the performance of the defended encoder. We observe that by adding more diverse training data, the performance can be improved for downstream tasks that share features with the private subset. The observation of increased performance is contrary to standard watermarking in supervised settings. An intuition to this behavior is that our defense does not directly influence the downstream task but only transforms the representations. In contrast, in supervised learning, watermarking directly impacts the task that the model is trained for, yielding performance loss.

To provide the defended encoder with a stronger unique signal for DSI, we train differently on the private seed than on the original training data. Specifically, we apply weaker augmentations to the private data points. This enables the defended encoder to overfit more easily to the private seed. To this end, we modify the training procedure of the DINO framework (Caron et al., 2021). Concretely, we change the standard DINO's data pre-processing and increase the size of crops for images passed through the student and teacher encoders. This is applied only to data points from the private subset and not to the original training points. Instead of using the student's crops that cover only small areas (less than 50%) of an input image, we increase the size of crops to the range between 70% and 90%. Furthermore, we also increase the size of the crops for the teacher encoder from greater than 50% to greater than 90%. DSI leverages this property and detects different behavior of encoders on their private training data versus unseen test data. Next, in our empirical evaluation we show that the unique signature from the private data seed is transferred to stolen encoder copies, making DSI applicable to perform ownership resolution. To further amplify the signal from the training signature, we add the victim's projection head on top of the verified encoders.

## 4.2 Watermarked Sentence-Embedding Encoders

We find that DSI (as well as standard DI) do not yield significant results in detecting ownership for stolen language encoders. This happens because of multiple reasons. First, all victims, stolen copies, and independent encoders use the same underlying pre-trained transformers. This similarity overlies the individual signals from datasets used for fine-tuning to sentence embeddings. Second, victim encoders are usually fine-tuned over a small number of epochs (10 for TinyBERT and 3 for BERT and RoBERTa). We observe that with longer fine-tuning, the results returned by DI on *victim encoders* become more confident. However, the signal on the *stolen encoders* remains insignificant.

As an alternative approach, we develop a new watermarking-based defense against stealing sentence embedding encoders. We embed the watermark starting from an already fine-tuned encoder. This is a realistic scenario where the model owner would like to add a watermark post hoc at a low cost. To embed the watermark, we perform a few iterations of training, where we always alternate between one iteration of the original sentence embedding training (with SimCSE), and then one iteration of training for a downstream task. During training of the downstream task, we add an additional fully-connected layer, which serves as our secret key during verification. Our watermarking approach is visualized in Figure 1b. In this work, we select SST2 (binary classification for sentiment analysis) as the watermark downstream task. Note, however, that a defender can select from many possible downstream tasks, reshuffle or flip the labels, or use their own private downstream task, which makes the detection of the watermark much harder.

To resolve ownership, a verifier simply attaches the fully-connected layer (secret key) to the output of an encoder suspected to be a stolen copy. Then, agreement between the outputs of the victim encoder (plus fully-connected layer), and the outputs of the potentially stolen copy (plus fully-connected layer) on the secret downstream task is measured as the percentage of labels where both outputs agree. We resolve that an encoder was stolen if the agreement is above the threshold of 95%, otherwise, the encoder is marked as independent. We assume that the adversary does not obfuscate the output representations as described in Dziedzic et al. (2022b).

## 5 EMPIRICAL EVALUATION

We evaluate our methods for stealing and defending transformer-based encoders trained on different vision and NLP datasets.

### 5.1 STEALING TRANSFORMERS

**Vision.** For ImageNet victims, we use the ViT's checkpoints released by the original DINO paper (Caron et al., 2021). For CIFAR10, we train Tiny ViTs from scratch. All training procedures follow (Caron et al., 2021) unless otherwise specified, using 300 epochs, a batch size of 256, and the learning rate is set to 5e-4 with a cosine annealing scheduler. For stealing, we experiment with different numbers of queries from various datasets, including CIFAR10, CIFAR100, SVHN, STL10, and ImageNet. Details on the datasets are included in Appendix B.1, while further insights on the training can be found in Appendix B.2. We also use ViTs to instantiate stolen encoders and experiment with different loss functions, such as MSE, InfoNCE (Chen et al., 2020), or SoftNN (Frosst et al., 2019). Unless otherwise specified, our stealing procedure follows Dziedzic et al. (2022a).

We assess the performance of victim and stolen encoders using standard linear evaluation (Chen et al., 2020) on four downstream tasks: CIFAR10, CIFAR100, SVHN, and STL10 (see Table 1). We observe that stealing with the victim's training distribution yields the closest performance of the stolen copy to the victim. The MSE loss performs better for the lower query regime (<100k) while contrastive losses like InfoNCE yield higher accuracy scores for the larger number of queries. When stealing with more complex datasets (*e.g.,* CIFAR10 vs SVHN) we can obtain a better generalization of the stolen copy, which is shown by higher accuracy on more downstream tasks. We also apply MixMatch for the case when an attacker has limited access to the API and wants to use as few queries as possible. Thus, in this case, we operate in the regime with a number of queries ranging from 4k to 10k, see Table 2. Our results show that MixMatch outperforms all other methods, even when provided with fewer labeled queries, and retains high similarity to the victim encoder when measured with the cosine similarity score between the representations from the victim encoder and a stolen copy.

**Language.** We steal from BERT-based sentence embedding encoders fine-tuned on nli-for-SimCSE (Gao et al., 2021) ("nli"), QQP (Iyer et al.) ("qqp"), and Flickr30k (Young et al., 2014) ("flickr"). For more details on the datasets and our pre-processing, see Appendix B.3. For TinyBERT, we fine-tune the victims using SimCSE (Gao et al., 2021), for Bert Base[3] ("BERT"), and RoBERTa Large[4] ("RoBERTa") we use models from Hugging Face as victim encoders. We fine-tune independent models using SimCSE and initialize our stolen copies with pretrained transformers of the victim architecture. Following SimCSE, we replace the original BERT pooling layer of our stolen models with a custom pooler that returns the [CLS] representation and add a randomly initialized MLP layer on top before stealing.[5] For stealing, we issue 60K queries, using sentences from our three datasets against the victim encoders and fine-tune our stolen copies with the resulting representations for 20 (TinyBERT and BERT), or 5 (RoBERTa) epochs, respectively, using MSE loss. Further details on our training and stealing methods are provided in Appendix B.4.

Table 3 depicts the performance of our victim and stolen encoders, evaluated on tasks from the SentEval benchmark. We observe that across all base encoders (TinyBERT, BERT, and RoBERTa), the performance of the stolen copies is comparable to their respective victim encoders over most benchmark tasks. This holds true even for the large qqp dataset (~2.6M training samples) and stolen

---

[3]https://huggingface.co/princeton-nlp/sup-simcse-bert-base-uncased
[4]https://huggingface.co/princeton-nlp/sup-simcse-roberta-large
[5]We also experiment with stolen copies initialized without the MLP layer and obtained similar performance in the benchmarks.

Table 1: **Performance of Vision Transformers.** We follow SimCLR (Chen et al., 2020) and do a linear evaluation of the encoders (denoted as EN) on downstream tasks. $f_v$ and $f_v'$ are victim encoders trained on data $D_v$, $f_s$ is the stolen encoder extracted using queries from a given stealing dataset $D_s$. $M$ is the type of the underlying encoder, where T is ViT Tiny, S is ViT Small, and B is ViT Base. Victim encoder with an asterisk (*) is a pre-trained encoder provided by Clarifai (2022). Other encoders are trained using the DINO code-base (https://github.com/facebook). CE denotes the Cross-Entropy loss and MSE is Mean Squared Error. For a given stealing dataset, we test loss functions: CE, MSE, InfoNCE, and SoftNN and report results for the best performing one.

| EN | M | Loss | # OF QUERIES | $D_v$ | $D_s$ | CIFAR10 | CIFAR100 | SVHN | STL10 |
|----|---|------|--------------|-------|-------|---------|----------|------|-------|
| $f_v$ | | - | - | SVHN | - | 73.3 | 49.03 | **90.4** | 36.2 |
| $f_v$ | T | - | - | CIFAR10 | - | **88.3** | 59.6 | 67.2 | **64.5** |
| $f_v'$ | | - | - | CIFAR10+SVHN | - | 87.9 | **59.97** | 78.4 | 63.3 |
| $f_s$ | T | MSE | | SVHN | SVHN | 62.6 | 38.1 | **88.8** | 33.1 |
| $f_s$ | T | MSE | 50K | SVHN | CIFAR10 | 72.7 | 49.3 | 56.8 | **37.2** |
| $f_s$ | T | MSE | | | CIFAR100 | **72.9** | **49.7** | 77.7 | 33.1 |
| $f_s$ | T | MSE | | CIFAR10 | SVHN | 60.1 | 35.0 | **67.2** | 31.3 |
| $f_s$ | T | MSE | 50K | | CIFAR10 | **86.9** | 58.7 | 64.7 | 52.7 |
| $f_s$ | T | MSE | | | CIFAR100 | 86.2 | **59.1** | 64.6 | **53.6** |
| $f_v$ | S | - | - | IMAGENET | - | 93.06 | 73.86 | 77.15 | 96.14 |
| $f_s$ | S | MSE | 50K | IMAGENET | CIFAR10 | **88.75** | **58.4** | 60.3 | **42.5** |
| $f_s$ | S | MSE | 100K | IMAGENET | SVHN | 62.04 | 35.0 | **77.1** | 38.8 |
| $f_v$ | S | - | - | CIFAR10 | - | 87.92 | 57.14 | 72.28 | 76.41 |
| $f_v$ | S | - | - | CIFAR10+SVHN | - | **88.07** | **57.49** | **78.07** | 76.28 |
| $f_v$ | S | - | - | CIFAR10+GTSRB | - | 87.67 | 56.62 | 69.97 | 74.39 |
| $f_v$ | S | - | - | CIFAR10+FASHIONMNIST | - | 88.05 | 57.19 | 71.74 | **76.83** |
| $f_v$ | B | - | - | IMAGENET | - | 93.8 | 78.8 | 67.1 | 98.9 |
| $f_s$ | B | MSE | 50K | CIFAR10 | CIFAR10 | 85.49 | 49.49 | 54.6 | 71.3 |

Table 2: **MixMatch for Stealing from ViT Tiny.** We compare the MixMatch semi-supervised method of training a stolen encoder with previous approaches. We report the accuracy for CIFAR10 (and for SVHN in Table 17). We tune the temperature parameter for sharpening in MixMatch and report the highest results for a downstream task per a given query number. We use the same number of unlabeled examples for MixMatch as the number of stealing queries. The stealing methods use different loss functions. CE denotes Cross-Entropy, MSE is the Mean Squared Error, SoftNN is Soft-Nearest Neighbor, and we also use the InfoNCE loss. $C(\cdot, f_v)$ (Cosine Similarity Score) is between the victim model and the stolen copy, following Dziedzic et al. (2022b).

| | | | Number of queries | | | | | | | | |
|-|-|-|-|-|-|-|-|-|-|-|-|
| | | | | 4K | | | 5K | | | 10K | |
| Stealing Method | Victim Data | Stealing Data | CIFAR10 | SVHN | $C(\cdot, f_v)$ | CIFAR10 | SVHN | $C(\cdot, f_v)$ | CIFAR10 | SVHN | $C(\cdot, f_v)$ |
| CE | | | 65.93 | 40.77 | 0.90 | 68.96 | 46.23 | 0.92 | 76.11 | 51.01 | 0.97 |
| MSE | | | 68.52 | 44.09 | 0.92 | 70.62 | 48.94 | 0.94 | 77.12 | 50.00 | 0.98 |
| SoftNN | CIFAR10 | CIFAR10 | 64.05 | 43.52 | 0.84 | 66.83 | 47.02 | 0.86 | 74.14 | 48.95 | 0.92 |
| InfoNCE | | | 66.46 | 44.37 | 0.87 | 68.51 | 48.96 | 0.89 | 75.59 | 48.15 | 0.94 |
| **MixMatch** | | | **72.86** | **50.31** | 0.91 | **75.04** | **50.9** | 0.93 | **78.1** | **51.5** | 0.95 |

copies obtained with only 60K queries, *i.e.,* ~40x fewer queries than training samples. In general, the performance of encoders stolen with nli and qqp is higher than the one of encoders stolen with flickr. We suspect this is due to the low semantic diversity in flickr which consists only of 30K images with five semantically equal captions each, leading to semantic overlap within the 60K stealing-queries.

We further explore the impact of the number of stealing queries on the performance of the stolen copies. Our results in Table 9 in Appendix C.1 highlight a performance decrease when reducing the number of stealing queries. The performance drop is most significant between 10k and 20k queries. This motivates an evaluation of the effectiveness of our method to re-use representations for semantically similar sentences in this setup. We query the stolen model copy with 10k sentences from the nli dataset and assign the obtained representation also to the semantically equal positive partner of each sentence. This results in 20k fine-tuning samples for the stolen copy. Our results in Table 4 highlight that the performance of the stolen copy with only 10k queries (augmented to 20k data points) is similar to the original stealing with 20k sentences.

Table 3: **Performance of NLP Transformers.** We follow SimCSE (Gao et al., 2021) and use the SentEval benchmark. $f_v$ denotes the victim encoder trained on data $D_v$. $f_s$ is the stolen encoder extracted using queries from a given stealing dataset $D_s$. For stealing, we use 60,000 queries to the victim encoder and fine-tune our stolen copy with the resulting outputs for 20 (TinyBERT (T) and BERT (B)), and 5 (RoBERTa (R)) epochs. Victim encoders with an asterisk (*) are pre-trained encoders from Hugging Face obtained by the SimCSE code-base (`https://github.com/princeton-nlp/SimCSE`), while other encoders are trained using the SimCSE code-base.

| EN | | $D_v$ | $D_s$ | CR | MPQA | MR | MRPC | SST2 | SUBJ | TREC | Avg.$_{STS}$ | Avg.$_{Tran}$ | SICKR | STSB |
|---|---|---|---|---|---|---|---|---|---|---|---|---|---|---|
| $f_v$ | T | nli | - | **74.69** | **82.17** | **68.51** | **71.93** | **72.02** | **87.99** | **59.74** | **76.64** | **73.86** | **73.67** | **79.61** |
| | | qqp | - | 72.27 | 79.1 | 65.93 | 70.31 | 68.92 | 85.83 | 57.17 | 66.19 | 71.36 | 56.15 | 76.22 |
| | | flickr | - | 71.87 | 79.39 | 66.56 | 71.52 | 69.61 | 86.89 | 55.25 | 72.83 | 71.58 | 66.63 | 79.03 |
| $f_s$ | T | nli | nli | **73.56** | 79.67 | 67.16 | **71.05** | 71.33 | **87.28** | 56.53 | **73.48** | 72.37 | **69.56** | 77.40 |
| | | | qqp | 73.48 | **79.88** | **67.22** | 70.78 | **71.9** | 87.23 | 56.51 | 71.08 | **72.42** | 65.74 | 76.41 |
| | | | flickr | 71.88 | 78.12 | 66.91 | 70.02 | 70.64 | 87.12 | **57.45** | 71.71 | 71.73 | 65.98 | **77.44** |
| | | qqp | nli | 71.42 | 78.08 | 66.88 | 70.14 | 70.64 | 86.62 | 50.24 | 64.97 | 70.57 | 55.18 | **74.75** |
| | | | qqp | **72.2** | **78.23** | 67.07 | 71.05 | **71.67** | 86.83 | **52.97** | 64.36 | **71.43** | 54.66 | 74.06 |
| | | | flickr | 71.95 | 77.78 | **67.14** | **71.61** | **71.67** | **86.96** | 49.34 | **65.23** | 70.92 | **56.37** | 74.09 |
| | | flickr | nli | 70.51 | **78.53** | 66.66 | **71.69** | 71.67 | 85.7 | **54.35** | 70.53 | **71.30** | 63.43 | 77.63 |
| | | | qqp | **71.05** | 78.41 | 66.63 | 71.49 | 70.87 | **86.46** | 53.6 | 68.57 | 71.22 | 60.42 | 76.73 |
| | | | flickr | 70.66 | 77.19 | **66.74** | 71.12 | 71.22 | 86.13 | 53.76 | **70.92** | 70.97 | **63.82** | **78.03** |
| $f_v$* | B* | nli | - | 89.20 | 89.67 | 82.88 | 73.51 | 87.31 | 94.81 | 88.40 | 81.57 | 86.54 | 80.39 | 84.26 |
| $f_s$ | B | nli | nli | **89.05** | 88.95 | **80.49** | 74.98 | 86.35 | 93.39 | 66.73 | **81.45** | 82.85 | **79.84** | **83.07** |
| | | | qqp | 88.25 | **89.18** | 79.74 | **75.20** | **86.58** | **94.09** | **69.08** | 79.93 | **83.16** | 77.24 | 82.62 |
| | | | flickr | 82.01 | 88.50 | 74.99 | 72.77 | 82.11 | 91.77 | 63.41 | 79.23 | 79.37 | 76.96 | 81.50 |
| $f_v$* | R* | nli | - | 92.37 | 90.52 | 88.04 | 76.64 | 92.31 | 95.13 | 91.20 | 83.76 | 89.46 | 81.95 | 86.70 |
| $f_s$ | R | nli | nli | 92.00 | **90.72** | 86.36 | **76.41** | 91.76 | 94.19 | 86.00 | 82.15 | 88.21 | **81.03** | 85.21 |
| | | | qqp | **92.58** | 90.69 | **87.02** | 75.54 | **92.53** | **94.48** | **88.60** | **82.82** | **88.78** | 80.79 | **85.84** |
| | | | flickr | 91.74 | 90.14 | 85.13 | 74.72 | 90.39 | 93.13 | 83.20 | 79.61 | 86.92 | 79.06 | 82.55 |

Table 4: **Re-using Representations.** We assign the same extracted representation to a given query and its semantically similar sentences. # Samples denotes the final number of sentences used to fine-tune the stolen encoder.

| EN | | $D_v$ | $D_s$ | # Queries | # Samples | CR | MPQA | MR | MRPC | SST2 | SUBJ | TREC | Avg.$_{STS}$ | Avg.$_{Tran}$ | SICKR | STSB |
|---|---|---|---|---|---|---|---|---|---|---|---|---|---|---|---|---|
| $f_s$ | B* | nli | nli | 10000 | 10000 | 84.58 | 87.64 | 77.68 | 75.69 | 83.94 | **93.56** | **69.19** | 59.25 | 81.75 | 56.63 | 61.87 |
| | | | | **10000** | **20000** | **86.89** | **88.59** | **78.56** | **76.42** | 85.44 | 93.29 | 68.62 | 68.35 | 82.54 | 65.30 | 71.40 |
| | | | | 20000 | 20000 | 87.42 | 89.11 | 79.13 | 75.66 | 86.93 | 93.45 | 70.14 | 69.35 | 83.12 | 65.51 | 73.18 |

## 5.2 DEFENDING TRANSFORMERS

**DataSeed Inference.** We design DataSeed Inference to incorporate a private seed into the victim's training set. Concretely, we insert randomly-selected 10k images from the SVHN and GTSRB training splits into the CIFAR10 training sets. We train ViT Tiny and ViT Small on the combined datasets as victims. We use weaker augmentations for the private set, which results in more similar inputs during training and evaluation and provides a stronger signal for DSI. Otherwise, the training procedures for the victim and stolen encoders are the same as before. Table 12 shows that DSI is more effective when the projection head in training DINO uses shallower MLPs. This result meets our intuition: since the training loss is minimized directly with the projection head, some information about the training signature is lost when the head is removed. Therefore, to make the training signature easier to detect, we attach the victim's projection head on top of the verified models. We use 20k randomly-selected images from the rest of the training splits as the validation set in DSI. We train GMMs with 10 components across all settings. For more details, see Appendix B.2. The results in Table 5 demonstrate that DSI is able to differentiate the stolen transformers of CIFAR10 by injecting a private subset of SVHN and GTSRB, even when the stealing datasets are different from training. Similar to DI for supervised learning (Maini et al., 2021), the victim encoder typically has the largest $\Delta\mu$ and the smallest p-values. We also find that increasing the size of the private set makes this defense more effective. Finally, our comparisons with the independent models ensure that we avoid false positives when detecting stolen models (see last two rows in Table 5). We also assess the performance of the encoders trained with CIFAR10 and a private data seed. We observe that these encoders achieve similar performance to the victim trained only on CIFAR10 and can outperform it when assessed on the downstream tasks that share features with the private subset, see Table 1.

**Watermarking Language Encoders.** We present the performance of the watermarked encoders as well as the success of our watermarking in Table 6. We use the initial fine-tuned sentence embedding

Table 5: **DataSeed Inference for Vision Transformers.** We train ViT-Tiny (left two columns) and Vit-Small (right two columns) using DINO. We add 10k data points from SVHN and GTSRB into the CIFAR10 training set. $f_v$ is the victim transformer trained on data $D$, $f_s$ is a transformer stolen with queries from a dataset $D$, and $f_i$ is an independent encoder trained on data $D$ (different than the victim's private training data). Each value is an average over 3 trials. $\Delta\mu$ is the effect size from the statistical t-test. An encoder is marked as stolen if the p-value is smaller than a threshold of 0.05.

| | | ViT Tiny | | | | | ViT Small | | | | |
|---|---|---|---|---|---|---|---|---|---|---|---|
| Victim's data → | | *CIFAR10+SVHN* | | | *CIFAR10+GTSRB* | | | *CIFAR10+SVHN* | | *CIFAR10+GTSRB* | |
| Encoder | $D$ | p-value | $\Delta\mu$ | $D$ | p-value | $\Delta\mu$ | $D$ | p-value | $\Delta\mu$ | $D$ | p-value | $\Delta\mu$ |
| $f_v$ | CIFAR10+ SVHN | 8.34e-63 | 16.85 | CIFAR10+ GTSRB | 6.31e-19 | 8.81 | CIFAR10+ SVHN | 1.47e-262 | 35.29 | CIFAR10+ GTSRB | 4.95e-77 | 18.64 |
| $f_s$ | CIFAR100 | 5.69e-12 | 6.79 | CIFAR100 | 1.84e-4 | 3.56 | CIFAR100 | 5.31e-4 | 3.27 | CIFAR100 | 4.61e-3 | 2.06 |
| | STL10 | 2.54e-2 | 1.95 | STL10 | 2.79e-13 | 7.21 | STL10 | 2.27e-2 | 1.93 | STL10 | 5.53e-5 | 3.87 |
| | ImageNet | 6.80e-4 | 2.47 | ImageNet | 1.74e-5 | 4.14 | ImageNet | 3.24e-2 | 1.91 | ImageNet | 1.74e-3 | 2.92 |
| | CIFAR10 | 0.63 | -0.33 | CIFAR10 | 0.18 | 0.88 | CIFAR10 | 0.57 | -0.19 | CIFAR10 | 0.18 | 0.89 |
| $f_i$ | CIFAR10+ FashionMNIST | 0.88 | -1.17 | CIFAR10+ FashionMNIST | 0.42 | 0.20 | CIFAR10+ FashionMNIST | 0.95 | -1.64 | CIFAR10+ FashionMNIST | 0.94 | -1.61 |

Table 6: **Watermarking Sentence Embedding Encoders.** We embed the watermark into an encoder and present the performance of the downstream task, the underlying encoder, and the comparison between the victim $f_v$, stolen $f_s$, and independent $f_i$ encoders. *Steps* denotes the number of fine-tuning steps for watermarking. Agreement is denoted by $Agr$ and accuracy on the watermark downstream task by $Acc$. Both are given in %. Test loss is denoted as $L$, p-value as $p$, and effect size as $\Delta\mu$.

| Steps | $Agr(f_v, f_s)$ | $Agr(f_v, f_i)$ | $Acc(f_v)$ | $L(f_v)$ | STS | SICKR | STSB | $Acc(f_s)$ | $L(f_s)$ | $p(f_v, f_s)$ | $\Delta\mu(f_v, f_s)$ | $p(f_v, f_i)$ | $\Delta\mu(f_v, f_i)$ |
|---|---|---|---|---|---|---|---|---|---|---|---|---|---|
| 100 | 97.59 | 88.3 | 57 | 0.68 | 0.77 | 0.74 | 0.79 | 56.42 | 0.68 | 0.79 | 0.27 | 0.0016 | 3.15 |
| **200** | **96.44** | **67.54** | **65.37** | **0.65** | **0.76** | **0.73** | **0.79** | **64.79** | **0.65** | **0.64** | **0.47** | **1.74E-14** | **7.73** |
| 300 | 97.71 | 72.59 | 65.37 | 0.63 | 0.76 | 0.73 | 0.8 | 65.14 | 0.63 | 0.62 | 0.5 | 9.87E-22 | 9.71 |
| 400 | 96.56 | 59.98 | 69.95 | 0.61 | 0.76 | 0.73 | 0.79 | 70.18 | 0.61 | 0.49 | 0.69 | 1.43E-27 | 11.07 |
| 500 | 96.33 | 60.66 | 71.1 | 0.59 | 0.75 | 0.71 | 0.79 | 71.1 | 0.6 | 0.46 | 0.74 | 2.67E-28 | 11.24 |
| 600 | 96.79 | 62.73 | 71.56 | 0.59 | 0.74 | 0.69 | 0.78 | 72.02 | 0.59 | 0.44 | 0.78 | 4.65E-32 | 12.03 |
| 700 | 96.44 | 62.61 | 71.44 | 0.58 | 0.72 | 0.67 | 0.77 | 72.25 | 0.58 | 0.45 | 0.76 | 2.30E-35 | 12.69 |
| 800 | 96.79 | 56.31 | 73.85 | 0.56 | 0.71 | 0.66 | 0.76 | 72.71 | 0.57 | 0.45 | 0.76 | 2.91E-42 | 14 |
| 900 | 96.34 | 59.75 | 73.39 | 0.55 | 0.71 | 0.65 | 0.76 | 72.13 | 0.56 | 0.42 | 0.8 | 3.02E-45 | 14.53 |

encoder as the independent encoder. This can be considered the worst-case evaluation because the representations returned from this encoder are expected to be most similar to its watermarked derivation. We show that even in this worst-case, the independent model is never incorrectly resolved as being a stolen copy. To compute the p-values, we leverage the confidence scores (softmax outputs) for the correct labels from the downstream task and use the t-test. The p-values indicate that there is a significant difference between the distribution of the confidence scores from independent vs victim encoders (p-value < 5%). In contrast, the difference is not significant between the victim and stolen encoders. Our results highlight that a relatively small number of fine-tuning steps (*e.g.,* 200 alternations between the original task and downstream task) are sufficient to successfully embed the watermark into the encoder while preserving the high performance of the defended encoder on other unrelated and general downstream tasks.

# 6 CONCLUSIONS AND FUTURE WORK

Modern APIs offer access to high-value transformer-based encoders for generating representations of given input text or images. We demonstrate how to steal these transformers in the language and vision domains by using only representations. Our stealing requires up to 40x fewer queries than the number of training data points used to train the victim and it yields stolen copies with comparable performance on standard benchmarks. We further decrease the number of stealing queries by using semantically similar sentences in the language domain and semi-supervised learning in the vision domain. We propose a new defense for vision transformers, where a private data seed acts as the training signature of the victim encoder. We find that this type of defense is unable to protect sentence embedding encoders since the signal of the private data subset used during fine-tuning is overlaid with the data used during pre-training of the base encoder. To overcome this obstacle, we propose a method for embedding a watermark into the encoders by fine-tuning a defended encoder with a specific downstream task.

## 7 ETHICS STATEMENT

Even though we place our work in a public access setting and implement our stealing based on representations such as the ones returned by public APIs, we do not actually steal the models from real-world APIs. All the models stolen in this work are either trained by ourselves or obtained from open-source platforms, such as Hugging Face. However, we note that several public APIs expose access to the same or similar models. Hence, our attacks could potentially be applied against such APIs to steal their models. Given that the models attacked in this work are publicly available anyways, their extraction from an API would not cause additional harm to the API provider. We still decided to keep the code implementing our model stealing confidential to minimize the risk of exposure through our work for such providers. Instead, we submit only the code implementing our defenses.

## 8 REPRODUCIBILITY STATEMENT

Due to the reasons elaborated in the ethics statement, we do not publicly submit the code of our stealing procedure. Instead, for reproducibility, we describe our approach in detail—documenting the frameworks we used, all hyperparameters for training, and the architectures. We submit the code for our defense in the supplementary material. In the *README.md* file we provide the main commands needed to run our code. We also describe the pointers to the crucial parts of the code that directly reflect the implementation described in the main part of the submission. We also describe the defenses used in our experiments thoroughly.

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

# A    ADDITIONAL RELATED WORK

**Self-Supervised Learning.**    In computer vision, one of the most popular self-supervised algorithms is contrastive learning (Chen et al., 2020; van den Oord et al., 2018), where representations that come from differently transformed views of the same image are brought closer to each other and the representations from views of different inputs are repelled. In NLP, a popular self-supervised pre-training approach is to mask selected words in the input sequence and train the model to predict that masked words Devlin et al. (2018).

**Transformers.**    Transformer (Vaswani et al., 2017) is becoming a ubiquitous architecture in NLP and computer vision. While the original transformer consists of an encoder and decoder component, our work only studies the encoder part for representation learning. Transformers are composed of several identical layers, namely a multi-head attention sublayer followed by a feed forward sublayer. The multi-head attention sublayer utilizes the self-attention mechanism to learn the pairwise relationships between all tokens. Self-attention is the key to the success of transformers, which makes parallel training and learning long-range dependency between tokens easier.

**NLP transformers.**    In this paper, we investigate BERT-based models (Devlin et al., 2018), pre-trained bidirectional transformers (Vaswani et al., 2017). In addition to BERT Base, we also use TinyBERT (Jiao et al., 2019), obtained by distilling BERT to a smaller transformer architecture, and RoBERTa Large (Liu et al., 2019), which is pre-trained on a larger dataset than BERT. Then we analyze the task of learning highly generic sentence representation, a fundamental problem in NLP (Kiros et al., 2015; Hill et al., 2016; Logeswaran & Lee, 2018).

**Vision Transformers.**    Vision transformers (Dosovitskiy et al., 2020; Touvron et al., 2021; Park & Kim, 2022), inspired by transformer models from the NLP domain, achieve state-of-the-art performance for many downstream tasks, such as standard image classification (He et al., 2022) or segmentation (Caron et al., 2021). They are based on either self-distillation with no labels (Caron et al., 2021) or masked autoencoders (**MAE**) (Devlin et al., 2018; Dosovitskiy et al., 2020). **MAE** is a masked autoencoder for the self-supervised training of vision transformers. The approach is based on training an asymmetric encoder-decoder architecture that learns to reconstruct randomly masked patches of an input image. **BEiT** (Bao et al., 2022) is another similar approach to MAE that also masks patches of an image and learns to recover the original inputs.

**Watermarking Encoders**    The watermark scheme proposed by Dziedzic et al. (2022a) trains the defended encoder simultaneously with an augmentation predictor. As the watermarking task, the defended encoder is trained to predict the rotation range of an input image. For watermark verification, an encoder's representations are evaluated on the augmentation predictor to obtain the accuracy on the watermark task and when the accuracy is significantly higher than 50%, the encoder is marked as stolen. SSLGuard (Cong et al., 2022) is another watermarking-based defense. First, it generates a secret key, a verification dataset, and a decoder. A watermarked encoder and its stolen copies map samples in the verification dataset to secret representations that in turn can be transformed into the secret key through the decoder. For independently trained encoders, the decoder transforms the representations generated from the verification dataset into random vectors. SSLGuard assumes a specific attack to which the trigger is transferred, which makes it vulnerable to new types of attacks. On the other hand, our DSI uses a private seed as a signature and does not rely on any assumptions about a type of stealing attack. Wu et al. (2022) manipulates the verification samples to obtain for them unique feature representations from protected and stolen encoders, which leads to unique predicted labels for any downstream task. However, it was shown that representations can be easily obfuscated (Dziedzic et al., 2022b) to remove their similarity to a stolen encoder.

**Stealing from NLP Classifiers**    The stealing methods proposed in Chen et al. (2021) and Rafi et al. (2022) are for downstream classification tasks while we steal representations from self-supervised encoders in both language as well as vision domains. For the vision encoders, we consider a more challenging task and start stealing from randomly initialized encoders instead of pre-trained ones. Apart from improving the stealing process by leveraging semantically similar sentences, we also propose a defense method against stealing sentence embedding encoders using a watermark-based approach. Chen et al. (2021) steal fine-tuned classification models, while our extraction targets

encoders, hence, it is more general. Rafi et al. (2022) extract architectures and weights, while our attack method extracts representations. Additionally, Rafi et al. (2022) require adversaries to have more knowledge about the victim than is assumed in our work. For example, architecture extractions require access to the target machine.

# B  DETAILS ON THE EXPERIMENTAL SETUP

## B.1  VISION DATASETS

**MNIST** (Deng, 2012) The MNIST database of handwritten digits has a training set of 60,000 examples, and a test set of 10,000 examples. The digits have been size-normalized and centered in a fixed-size image. It consists of 28x28 black and white images with 10 classes.

**CIFAR10** (Krizhevsky, 2009): The CIFAR10 dataset consists of 32x32 colored images with 10 classes. There are 50000 training images and 10000 test images.

**CIFAR100** (Krizhevsky, 2009): The CIFAR100 dataset consists of 32x32 coloured images with 100 classes. There are 50000 training images and 10000 test images.

**SVHN** (Netzer et al., 2011): The SVHN dataset contains 32x32 coloured images with 10 classes. There are roughly 73000 training images, 26000 test images and 530000 "extra" images.

**ImageNet**(Deng et al., 2009): Larger sized coloured images with 1000 classes. As is commonly done, we resize all images to be of size 224x224. There are approximately 1 million training images and 50000 test images.

**STL10** (Coates et al., 2011): The STL10 dataset contains 96x96 coloured images with 10 classes. There are 5000 training images, 8000 test images, and 100000 unlabeled images.

## B.2  EXPERIMENTAL SETUP FOR STEALING VISION TRANSFORMERS

For vision, ViT Tiny, Small architectures are used for the CIFAR10 and SVHN victim encoders. We use DINO (Caron et al., 2021) to train ViT with patch size 16 in a self-supervised way. We resize the images to $224 \times 224$ so that we can use the hyperparameters provided in the DINO paper. We use 3-layer MLP and then apply an L2 normalization, and a weight normalized fully-connected layer as the DINO paper by default, and do experiments with the different number of MLP layers in **??**.

We train all victim models for 300 epochs. To train the stolen encoders, we used 150 epochs. We use AdamW optimizer and Cosine Annealing scheduler when training victim and stolen models. When training the victim model, the initial learning rate is 4e-5 while for training stolen models, the initial learning rate is 1e-3. A batch size of 128 or 256 was used for training the models. The temperature we use for InfoNCE is 0.07, and the temperature for soft nearest neighbors loss is 1000.

For the stealing experiments, we use the last 4 block outputs of dummy representations, in the same way as in DINO (Caron et al., 2021). Thus, the dimensions of output representations for ViT-tiny and ViT-small are 768 and 1536, respectively.

**Victim** We train tiny-ViTs on CIFAR10. All training procedures follow (Caron et al., 2021) unless otherwise specified. The number of epochs is 300, the $bs$=256 and $lr$=5e-4 with a cosine annealing scheduler. The summary of the victim encoder's performance can be found in Table 1.

**Stolen** When stealing from the victim encoders, we experiment with different numbers of queries from various datasets, including CIFAR10, CIFAR100, SVHN, and STL10. We also use both ResNet and TinyViT as the architectures for stolen encoders. Stolen encoders are trained by minimizing different loss functions, such as MSE, InfoNSE (Chen et al., 2020), or SoftNN (Frosst et al., 2019). Unless otherwise specified, the stealing procedure follows Dziedzic et al. (2022a).

**Independent** We use ViTs trained on other datasets than the victim's training set as independent encoders.

**Private Data Subsets** For DSI, we insert 5k, 10k SVHN, and 10k MNIST images into the CIFAR10 training sets respectively. The training procedures for the victim and stolen encoders are the same as before.

## B.3 NLP Datasets and Processing

**nli.** We use (Gao et al., 2021)'s nli-for-SimSCE dataset, consisting of 275,602 data rows from SNLI (Bowman et al., 2015) and MNLI (Williams et al., 2018). Each row holds three sentences, an original sentence, a positive entailment, and a contradiction. In training, the contradiction acts as a hard-negative.

**qqp.** We use the merve/qqp dataset from Hugging Face. The train split consists of 2,607,949 data rows, each holding two semantically equal questions. We use this data for training as positive pairs.

**flickr.** The flickr dataset consists of images, each annotated with five human-written captions. Following (Gao et al., 2021), we consider any two captions of the same image as a positive pair. We split the training and test set to 90%, 10%, making sure that all caption pairs related to one image end up in the same set. This yields 286,050 positive-pair training examples. When using flickr as a dataset for stealing, we drop the duplicates arising from generating all possible caption pairs before sampling the stealing-queries. Thereby, we mitigate a too small diversity over the stealing due to repeated queries.

## B.4 Experimental Setup for Stealing Language Transformers

**Victim.** As victim encoders, we use TinyBERT-based encoders and fine-tune them for the sentence embedding task on nli, qqp, and flickr by using SimSCE (Gao et al., 2021). For more details on data pre-processing and the datasets, see Appendix B.3. We fine-tune our encoders for 10 epochs, with batch size ($bs$)=128, learning rate ($lr$)=5e-5, and temperature=0.05. Additionally, we use transformers fine-tuned with nli from BERT [6], and RoBERTa [7] from Hugging Face as victim encoders. For fine-tuning and stealing over all encoders, we set the maximal input sequence length to 32 and use truncation and padding. The performance of our victim encoders can be found in Table 3.

**Stolen.** We initialize our stolen encoders with pre-trained transformers from Huggingface (prajjwal1/bert-tiny, bert-base-uncased, and roberta-large), in accordance with the respective victim encoder. During stealing, TinyBERT and BERT use an $lr$=1e-5, $bs$=256, and linear $lr$-scheduling with patience 200 iterations and factor 0.5. For RoBERTa, we use the same setup, however with bs=64, $lr$-patience 600 iterations, and $lr$=5e-6 when stealing with nli or flickr. As in SimCSE, we evaluate our stolen encoders on the SentEval benchmark.

**Independent.** We fine-tune the independent TinyBERT-based encoders on nli, qqp, and flickr in the same setup as the victim encoders. To obtain independent encoders based on BERT and RoBERTa, we fine-tune the respective base encoders on nli, qqp, and flickr for using SimCSE. We keep $lr$=5e-5, temperature=0.05, $bs$=128 and $bs$=32 for BERT and RoBERTa, respectively, but following (Gao et al., 2021), we fine-tune only for 3 epochs.

**DSI.** To evaluate DSI, we generate training datasets with a private dataset as secret seeds. We generate three such sets: 1) 50K flickr data points into the qqp dataset, 2) the full flickr, *i.e.,* $\sim$300K into qqp, and 3) 50K qqp into flickr. We fine-tune sentence embedding encoders based on TinyBERT using the same procedure as for the victim and independent encoders. When analyzing the performance of encoders trained with the data subsets in Table 7 and note that for most tasks the flickr+qqp encoder overpasses the performance of the encoder trained only on flickr, while the performance of qqp+flickr increases only for the STS benchmarks.

Table 7: **Performance of NLP Transformers with Mixed Dataset.** We follow SimCSE (Gao et al., 2021) and use the SentEval benchmark. $f_v$ denotes the victim encoder trained on data $D_v$. $f_s$ is the stolen encoder extracted using queries from a given stealing dataset $D_s$.

| EN | | $D_v$ | $D_s$ | CR | MPQA | MR | MRPC | SST2 | SUBJ | TREC | Avg.$_{STS}$ | Avg.$_{Tran}$ | SICKR | STSB |
|---|---|---|---|---|---|---|---|---|---|---|---|---|---|---|
| | | qqp+flickr(50k) | - | 70.67 | 76.71 | 64.74 | 70.51 | 67.2 | 83.96 | 55.15 | 70.13 | 69.85 | 62.26 | 78.01 |
| $f_v$ | T | qqp+flickr(300k) | - | 71.63 | 76.81 | 65.32 | 70.24 | 68.81 | 84.6 | **57.04** | **73.6** | 70.64 | 66.35 | **80.84** |
| | | flickr+qqp(50k) | - | **72.86** | **79.55** | **66.54** | **71.17** | 69.5 | **86.4** | 55.85 | 73.46 | **71.7** | **67.02** | 79.91 |

---

[6] https://huggingface.co/princeton-nlp/sup-simcse-bert-base-uncased
[7] https://huggingface.co/princeton-nlp/sup-simcse-roberta-large

Table 8: **Cosine Similarity Scores.** Scores calculated over 20K data points between a victim encoder $f_v$, copies $f_s$ stolen with data $D_s$, and independent encoders $f_i$ trained on $D_i$. Results reported with standard deviation. In case $f_v$ and $f_i$ use the same data, we train two independent encoders with the same architecture, data, and hyperparameters (gray). Results with a double asterisk (**) indicate that we compare two independently trained BERT or RoBERTa-based encoders, instead of comparing the public Hugging Face encoder (*) with an independently trained encoder. This serves for better comparability to results obtained with TinyBERT. The notation follows Table 3. For encoders trained with the additional private subset, stolen copies should be compared with independent encoders trained on the same public data without the inserted subset (blue).

| | | $f_s$ stolen with $D_s$ | | | $f_i$ trained on $D_i$ | | |
|---|---|---|---|---|---|---|---|
| | $f_v$ on $D_v$ | nli | qqp | flickr | nli | qqp | flickr |
| T | nli | 0.94±0.02 | 0.87±0.05 | 0.86±0.12 | 0.99±0.0 | 0.58±0.09 | 0.73±0.09 |
| | qqp | 0.75±0.08 | 0.83±0.04 | 0.66±0.1 | 0.61±0.08 | 0.95±0.01 | 0.52±0.08 |
| | flickr | 0.91±0.04 | 0.81±0.07 | 0.94±0.03 | 0.77±0.06 | 0.46±0.09 | 0.98±0.0 |
| | qqp+flickr | 0.81±0.04 | 0.37±0.09 | 0.84±0.05 | 0.5±0.09 | 0.69± 0.08 | 0.38±0.12 |
| | qqp+flickr(full) | 0.79±0.05 | 0.3±0.09 | 0.84±0.05 | 0.62±0.08 | 0.6± 0.09 | 0.59±0.1 |
| | flickr+qqp | 0.8±0.05 | 0.89±0.03 | 0.45±0.11 | 0.74±0.06 | 0.52±0.09 | 0.74±0.06 |
| B* | nli | 0.87±0.04 | 0.68±0.09 | 0.7±0.21 | 0.99±0.0** | 0.66±0.07** | 0.7±0.1** |
| R* | nli | 0.81±0.06 | 0.68±0.08 | 0.62±0.22 | 1.0±0.00** | 0.3±0.09** | 0.4±0.01** |

## C  ADDITIONAL EXPERIMENTAL RESULTS

### C.1  ANALYSIS OF THE NLP ENCODERS AND THEIR SIMILARITIES

As an alternative to the DSI, which is ineffective in protecting our sentence embedding encoders, we evaluate the similarity between the victim, stolen, and independent encoders, we use the cosine similarity score, as in (Dziedzic et al., 2022b). Note that the cosine similarity score should not be used for the ownership resolution. This is a very fragile metric and any deviation in the training of the stolen copy from the victim increases the distances between their representations. As shown in Dziedzic et al. (2022b), most obfuscations of stolen representations leave a stolen copy undetectable by the cosine similarity score. We track the cosine similarity between representations of the stolen

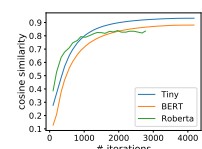

Figure 2: **Cosine Similarity Score over Stealing.**

and victim encoder over the course of stealing on 50 representations from the test dataset (Figure 2) and observe that similarity increases from the very beginning of stealing providing evidence of the highly accurate stealing process. Table 8 depicts the final cosine similarity scores between language encoders, stolen copies, and independent models after stealing is completed. We find that the stolen copies have higher scores than independent encoders trained on *different* datasets than the victim's data.

In Table 8, we also evaluate the cosine similarity of our encoders trained with a private seed against their stolen copies and independent encoders. We evaluate the similarity of our encoders trained with private subsets against their stolen copies and independent encoders. We evaluate qqp and flickr in both combinations as public and private data and do not consider nli. This is because the nli dataset contains SNLI, which, in turn, contains data from flickr which might corrupt our results. Experimenting with the full flickr dataset inserted into qqp serves a fair comparison in terms of the fraction of subset data, given that qqp holds roughly 10x more data than flickr (2.6M vs. 300k). Our results highlight that inserting a large enough subset of flickr increases the cosine similarity between the qqp+flickr encoder to an independent flickr encoder while it decreases the similarity to an independent flickr encoder. The same holds for the qqp-subset. By inserting larger fractions of flickr, the qqp+flickr encoder's representations also resemble more the ones of the nli encoder, which can be explained by nli containing some sentences from flickr. For meaningful ownership resolution, the similarity between the victim and its stolen copies has to be compared to the similarity of the victim and an independent encoder trained on the same public dataset but without the private subset (color-coded in blue in Table 8). Our results show that if a victim is stolen with the private data subset or an independent dataset, the similarity to its stolen copies is higher than to the independent encoder. However, the information on the private data subset is not transferred to the stolen copies when the stealing is done with the victim's original training data distribution. In a public setting, this

is problematic since APIs reveal information about types of data used to pre-train and fine-tune their encoders.

Note that for the pre-trained encoders, BERT* and RoBERTa* from Hugging Face, the cosine similarity scores to our independently trained encoders are close to zero. This might be caused by different factors, such as modifications to the code base or differences in the execution environment. Moreover, (Dziedzic et al., 2022b) showed that even a small obfuscation of the representations decreases the cosine similarity between two encoders substantially. Note that despite some inconsistencies in representations between BERT and RoBERTa from Hugging Face and our independently trained encoders, we are able to steal the encoders from Hugging Face. Our stolen copies do not only achieve high performance on benchmarks (Table 3) but also produce representations with high cosine similarity scores to the pre-trained encoders (Table 8). This highlights that while language transformers can be stolen effectively, the signal in their representations does not easily benefit ownership resolution.

Therefore, to evaluate a worst-case scenario (the exact data and training procedure of the victim are public and the independent encoder is trained in exactly the same way), we train BERT and RoBERTa-based encoders on nli twice and report the cosine similarity scores between these two encoders, instead. The encoders trained independently on the same data and with the same procedure yield a similarity score close to one (color-coded in gray)—higher than the victim and its stolen copies. This indicates that the scores cannot be used for ownership resolution since an honest party who has trained an encoder using the same data and procedure as the victim would be falsely accused of theft. These results are expected. In fact, standard DI cannot distinguish between a victim encoder and an independent encoder trained with the same procedure using the same data. Therefore, DI assumes that the victim's training data is private, which, as we motivated above might not be a realistic assumption for transformers trained on public data. This motivates our novel DSI.

Our experiments with DI, DSI and cosine similarity for language transformers, yield some additional interesting insights that need to be taken into account when designing future defenses against stealing NLP encoders. We observe that if the training and test sets of an encoder consist of semantically equal or similar sentences, DI is not even enable to mark a victim encoder. This is caused since the encoders are trained to output similar/same embeddings for semantically equal sentences. For example, when splitting the flickr dataset such that captions of the same image are distributed over the train and test sets, the distributions of the resulting representations are so similar that DI cannot extract any signal. Therefore, when attempting to identify stolen encoder copies in NLP, auditors need to be very careful not to accuse honest third parties of theft, when they hold semantically equal training data for their encoders than a victim encoder. Furthermore, we observe that the data pre-processing and tokenization in NLP plays a significant role for DI. While DI is able to mark victim encoders if their input data is batched and tokenized in exactly the same way, it is unable to do so when the data is tokenized differently (*e.g.,* with different padding, truncation, batching). This observation is intuitive since DI relies on an exact training signature. In ownership resolution, this, furthermore, does not represent a limitation since the owner of the victim knows the exact pre-processing method used during their training / fine-tuning. We also experimented with DI using only sentences above or below a certain length for ownership resolution at inference time. This implements the intuition that some very short or long sentences might represent a more unique training signature, however, neither DI nor DSI were able to successfully mark stolen copies. Finally, we observed that in nli (in contrast to qqp and flickr), DI was not even always able to mark the victim model. Results varied broadly, depending on the data sampled from the training and test set to fit the GMMs and estimate likelihood. We assume that this is due to nli containing data from many different domains (the underlying MNLI contains text from, among others, fiction, telephone speech, and letters). Subsampling might, hence, lead to having data from completely different distributions in training and test data and, thereby, prevent DI.

## C.2 NLP: KNOWN ARCHITECTURE BUT UNKNOWN CHECKPOINT

We observe that when stealing is started from a different checkpoint for the underlying encoder than the one used by the victim, the performance changes depending on the difference between the pre-trained checkpoints (first 3 rows in Table 10). For example, while the difference between using bert-base-uncased and bert-base-cased as the starting point for stealing is relatively small when the

Table 9: **Number of Stealing Queries and Impact on Model Performance.** We follow Sim-CSE (Gao et al., 2021) and use the SentEval banchmark. $f_v$ denotes the victim encoder trained on data $D_v$. $f_s$ is the stolen encoder extracted using queries from a given stealing dataset $D_s$. For stealing, we use a different number of queries to the victim encoder and fine-tune our stolen copy with the resulting outputs for 20 epochs. We use the BERT-based model pre-trained and taken from Hugging Face obtained by the SimCSE code-base, marked with an asterisks * (`https://github.com/princeton-nlp/SimCSE`) and steal using the nli dataset.

| EN | $D_v, D_s$ | | # Queries | CR | MPQA | MR | MRPC | SST2 | SUBJ | TREC | Avg.$_{STS}$ | Avg.$_{Tran}$ | SICKR | STSB |
|---|---|---|---|---|---|---|---|---|---|---|---|---|---|---|
| | | | 60k | **89.05** | 88.95 | **80.49** | 74.98 | **86.35** | **93.39** | 66.73 | **81.45** | **82.85** | **79.84** | **83.07** |
| | | | 50k | 88.04 | 88.68 | 78.96 | 75.74 | 86.01 | 92.76 | 64.49 | 80.98 | 82.1 | 78.96 | 82.99 |
| | | | 40k | 87.83 | 88.29 | 78.87 | 75.96 | 86.12 | 92.78 | 64.91 | 80.67 | 82.11 | 78.99 | 82.36 |
| $f_s$ | B* | nli | 30k | 87.18 | 89.13 | 78.26 | **76.4** | 84.4 | 92.83 | 65.33 | 79.09 | 81.93 | 76.9 | 81.29 |
| | | | 20k | 86.41 | **89.16** | 78.45 | 75.44 | 84.4 | 92.98 | 71.37 | 74.39 | 82.60 | 71.93 | 76.84 |
| | | | 10k | 83.81 | 87.52 | 78.08 | 74.39 | 84.06 | 93.28 | **73.27** | 60.03 | 82.06 | 53.86 | 66.2 |
| | | | 5k | 78.96 | 83.77 | 76.04 | 72.74 | 84.29 | 92.86 | 69.24 | 48.18 | 79.7 | 44.0 | 52.36 |
| | | | 1k | 80.15 | 83.09 | 77.42 | 70.68 | 84.75 | 93.0 | 68.12 | 45.45 | 79.67 | 53.14 | 37.75 |

Table 10: **Base Checkpoints and Influence on Model Performance.** We follow SimCSE (Gao et al., 2021) and use the SentEval banchmark. $f_v$ denotes the victim encoder trained on data $D_v$. $f_s$ is the stolen encoder extracted using queries from a given stealing dataset $D_s$. For stealing, we 60,000 stealing queries to the victim encoder and fine-tune our stolen copy with the resulting outputs for 20 epochs. We use the BERT-based model pre-trained and taken from Hugging Face obtained by the SimCSE code-base, marked with an asterisks * (`https://github.com/princeton-nlp/SimCSE`) and steal using the nli dataset. While the original BERT was initialized with bert-base-uncased, we vary the weight initialization of the checkpoint used for stealing. We also check performance after 1 epoch of fine-tuning (1 epoch). We use two different tokenizers. The API uses their original tokenizer (bert-base-uncased, potentially unknown to attacker), while the attacker uses the tokenizer that corresponds to the checkpoint from which they load the weights.

| EN | $D_v$ | $D_s$ | Stealing Initialization | CR | MPQA | MR | MRPC | SST2 | SUBJ | TREC | Avg.$_{STS}$ | Avg.$_{Tran}$ | SICKR | STSB |
|---|---|---|---|---|---|---|---|---|---|---|---|---|---|---|
| $f_s$ | B* | nli | nli | bert-base-uncased (baseline 20 epochs) | **89.05** | **88.95** | **80.49** | 74.98 | **86.35** | **93.39** | **66.73** | **81.45** | **82.85** | **79.84** | **83.07** |
| | | | | bert-base-cased (20 epochs) | 87.36 | 88.93 | 77.94 | **76.35** | 85.55 | 92.04 | 66.45 | 77.88 | 82.08 | 75.84 | 79.91 |
| | | | | bert-base-multilingual-cased (20 epochs) | 70.36 | 69.73 | 60.09 | 69.72 | 64.91 | 79.14 | 49.5 | 58.62 | 66.20 | 57.93 | 59.31 |
| $f_s$ | B* | nli | nli | bert-base-uncased (1 epoch) | **75.06** | 81.68 | **73.12** | 71.37 | 83.94 | **91.95** | **62.51** | 48.67 | **77.09** | **52.9** | 44.44 |
| | | | | bert-base-cased (1 epoch) | 69.28 | **81.9** | 64.96 | 70.07 | 82.34 | 90.18 | 52.2 | **49.77** | 72.99 | 51.42 | **48.13** |
| | | | | bert-base-multilingual-cased (1 epoch) | 64.15 | 69.18 | 54.11 | 68.79 | 58.94 | 65.91 | 30.78 | 43.08 | 58.83 | 44.75 | 41.41 |

original model used bert-base-uncased, the performance difference when using bert-base-multilingual-case is larger.

Next, we show that the underlying encoder leaves distinct traces on the fine-tuned sentence embedding model. We analyze the performance of different checkpoints only after a single epoch of fine-tuning during which we use the stolen embeddings (last 3 rows in Table 10). This computationally inexpensive step allows attackers to quickly identify the best-performing checkpoint that they have access to and continue the fine-tuning only for the best checkpoint. Note that evaluating different model checkpoint initializations does not require obtaining additional representations from the victim model. Instead, the obtained representations can be reused over all checkpoints.

## C.3 DATASET INFERENCE VS DISTRIBUTION INFERENCE IN VISION

Table 11 show that DI is a robust detection method. We train two ResNet models on the equal partitions of the CIFAR10 train set. The result suggests that we are able to differentiate between two such encoders as being trained independently. We also run the same experiments for the original version of dataset models, where we train two classifiers on the equal splits of the CIFAR10 train set. We observe that both methods are robust in this setting. To apply GMMs to supervised models, we discard the last classification layer and use the representations from the last but one layer to distinguish between these two classifiers using GMMs. This shows that the dataset inference is not only a *distribution* inference but the intended *dataset* inference. We can use the method based on GMMs when the public API exposes access to the softmax values for the supervised models.

Table 11: **Dataset Inference.** We divide the CIFAR10 train set into two equal splits. Then, we train on each split. We test if such two encoders are marked as independent. Each value is an average of 3 trials. $\Delta\mu$ is the effect size from the statistical t-test. DI denotes the original Dataset Inference proposed by Maini et al. (2021). For Blind Walk (rand) the results fluctuate due to the randomness coming from the selected data points and the embedding generation method.

| | DATASET FOR | DATASET FOR TRAINING MODELS | | | |
| METHOD | DATASET INFERENCE | 1ST HALF OF CIFAR10 | | 2ND HALF OF CIFAR10 | |
| | | P-VALUE | $\Delta\mu$ | P-VALUE | $\Delta\mu$ |
|---|---|---|---|---|---|
| GMMS ON ENCODERS | 1ST HALF OF CIFAR10 | 7.43E-12 | 6.85 | 0.56 | 0.57 |
| | 2ND HALF OF CIFAR10 | 0.11 | 1.82 | 2.15E-7 | 5.18 |
| DI (MAINI ET AL., 2021) RAND 10 POINTS | 1ST HALF OF CIFAR10 | 9.61E-25 | 1.99 | 0.63 | 0.01 |
| | 2ND HALF OF CIFAR10 | 0.98 | -0.14 | 3.68E-23 | 3.99 |
| DI (MAINI ET AL., 2021) RAND 100 POINTS | 1ST HALF OF CIFAR10 | 2.92E-149 | 1.99 | 0.43 | 0.31 |
| | 2ND HALF OF CIFAR10 | 0.30 | 0.32 | 2.24E-203 | 1.99 |
| DI (MAINI ET AL., 2021) RAND 1000 POINTS | 1ST HALF OF CIFAR10 | 2.54E-254 | 1.57 | 0.53 | 0.14 |
| | 2ND HALF OF CIFAR10 | 0.44 | 0.21 | 7.32E-203 | 1.83 |
| DI (MAINI ET AL., 2021) MINGD 10 POINTS | 1ST HALF OF CIFAR10 | 2.35E-12 | 1.92 | 0.999 | -1.82 |
| | 2ND HALF OF CIFAR10 | 0.27 | 0.02 | 4.14E-8 | 1.87 |
| DI (MAINI ET AL., 2021) MINGD 100 POINTS | 1ST HALF OF CIFAR10 | 1.56E-44 | 1.84 | 0.048 | 0.20 |
| | 2ND HALF OF CIFAR10 | 0.34 | 0.27 | 4.82E-50 | 1.84 |
| DI (MAINI ET AL., 2021) MINGD 1000 POINTS | 1ST HALF OF CIFAR10 | 1.69E-104 | 1.98 | 0.999 | -1.23 |
| | 2ND HALF OF CIFAR10 | 0.74 | 0.12 | 3.21E-112 | 1.99 |
| GMMS ON SUPERVISED-SOFTMAX | 1ST HALF OF CIFAR10 | 3.17E-63 | 16.86 | 0.43 | 0.79 |
| | 2ND HALF OF CIFAR10 | 0.64 | 0.67 | 7.95E-24 | 10.08 |
| GMMS ON SUPERVISED-LOGITS | 1ST HALF OF CIFAR10 | 3.19E-178 | 28.85 | 0.44 | 0.77 |
| | 2ND HALF OF CIFAR10 | 0.18 | 0.85 | 1.02E-105 | 22.01 |
| GMMS ON SUPERVISED-REPRESENTATIONS | 1ST HALF OF CIFAR10 | 4.47E-189 | 29.75 | 0.69 | 0.48 |
| | 2ND HALF OF CIFAR10 | 0.33 | -0.96 | 1.94E-133 | 24.83 |

Table 12: **Effects of Projection Heads on DI.** We perform dataset inference for DINO trained on CIFAR10 with different projection heads. DINO uses a 3-layer MLP followed by $\ell_2$ normalization and a linear layer as the projection head during training, which is then discarded for evaluations. The first column denotes the number of layers for MLP in the projection head. The second column denotes whether the projection head is included in DI. For each set-up, we also vary the number of components for GMM. The bolded number indicates that DI successfully detects the victim.

| NUMBER OF | HEAD | NUMBER OF GMM COMPONENTS | | | | | | |
| MLP LAYERS | INCLUDED | 2 | 5 | 10 | 20 | 30 | 40 | 50 |
|---|---|---|---|---|---|---|---|---|
| 0 | NO | **4.72E-2** | **6.09-4** | **3.67E-5** | **7.82E-8** | **9.23E-11** | **1.73E-15** | **5.43E-20** |
| 1 | NO | 5.63E-2 | **3.31E-4** | **7.43E-6** | **6.34E-7** | **8.92E-8** | **7.32E-8** | **5.66E-9** |
| 2 | NO | 0.71 | 0.53 | 0.38 | 0.11 | **8.57E-3** | **2.34E-4** | **1.21E-4** |
| 3 | NO | 0.52 | 0.53 | 0.65 | 0.68 | 0.14 | 0.25 | 0.36 |
| | ONLY-1-LAYER MLP | 0.42 | 0.57 | 0.28 | 0.33 | 0.64 | 0.13 | 0.26 |
| | ONLY-2-LAYER MLP | 0.73 | 0.82 | 0.57 | 0.39 | 0.26 | **3.91E-2** | **4.62E-2** |
| | ONLY-3-LAYER MLP | 0.83 | 0.12 | **1.12E-2** | **1.91E-6** | **2.43E-12** | **4.56E-16** | **8.34E-21** |
| | YES (ALL) | **5.92E-6** | **3.75E-3** | **5.63E-14** | **5.01E-9** | **1.49E-11** | **1.28E-22** | **1.67E-56** |

## C.4 INFLUENCE OF DIFFERENT PROJECTION HEAD SETTINGS

We investigate the effect of the depth and the final-layer dimension of DINO's projection head on the model's performance on downstream tasks. Table 13 shows that having a deeper MLP is crucial to the model's quality.

## C.5 MORE EXPERIMENTS ON STEALING ViT SMALL

Table 14 shows the performance of stolen models using different datasets, query numbers and loss functions. The result suggests that using the same dataset as the pretrained dataset gives the best result.

## C.6 NUMBER OF PARAMETERS

We count the number of parameters for the base encoders and their corresponding heads and show the results in Table 15. The ViT-tiny has much fewer parameters than even the single projection head. ViT-small has a smaller number of parameters than the full projection head used in DINO. Only the

Table 13: **Influence of Projection Heads.** We train Vit Tiny from scratch using DINO on CIFAR10 with different configurations of projection head with respect to number of layers and output dimension(denoted as od), and evaluate them on different downstream tasks: CIFAR10, SVHN.

| Model | Pretrained Dataset | Method | CIFAR10 | SVHN |
|-------|--------------------|--------|---------|------|
| ViT-tiny | CIFAR10 | DINO(Head 3+1) | 88.3 | 67.2 |
| ViT-tiny | CIFAR10 | DINO(Head 1+1) | 86.7 | 58.2 |
| ViT-tiny | CIFAR10 | DINO(Head 1+1, od192) | 85.1 | 59.2 |
| ViT-tiny | CIFAR10 | DINO(Head 1+1, od576) | 83.9 | 58.8 |
| ViT-tiny | CIFAR10 | DINO(Head 1+1, od3840) | 84.4 | 58.6 |
| ViT-tiny | CIFAR10 | DINO(Head 0+1) | 86.7 | 58.8 |
| ViT-tiny | CIFAR10 | DINO(Head 0+1, od192) | 86.2 | 58.5 |
| ViT-tiny | CIFAR10 | DINO(Head 0+1, od4096) | 84.6 | 58.5 |

Table 14: **Stealing ViT Small.** The victim encoder is a ViT Small pretrained on ImageNet with the DINO training approach. We assess the encoder on CIFAR10.

| Stealing Dataset | Loss | # of Queries | CIFAR10 |
|------------------|------|--------------|---------|
| *Victim Model* | N/A | N/A | 91.91 |
| CIFAR10 | MSE | 50000 | 88.75 |
| CIFAR10 | InfoNCE | 50000 | 87.15 |
| CIFAR100 | MSE | 50000 | 85.65 |
| SVHN | MSE | 100000 | 62.04 |
| ImageNet | InforNCE | 100000 | 84.91 |
| ImageNet | MSE | 100000 | 71.87 |
| ImageNet | InfoNCE | 200000 | 87.65 |
| ImageNet | InfoNCE | 250000 | 88.86 |
| ImageNet | MSE | 250000 | 80.30 |
| ImageNet | SoftNN | 250000 | 79.69 |

ViT-base has a larger number of parameters than the full projection head from DINO. The number of parameters in the projection heads is different for various encoder types due to their different dimensionality of output representations.

## C.7 FINE-TUNING

In Table Table 16, we present the results after fine-tuning with the SVHN dataset on ViT Small/16 pre-trained on ImageNet.

## C.8 MIXMATCH FOR SVHN

We present the results on applying the MixMatch semi-supervised learning to the SVHN dataset in Table 17.

Table 15: **Number of Parameters** in base encoders (backbones) and projection heads.

| Encoder Type | Base Encoder | Head-1 | Head-1-2 | Head-1-2-3 |
|---|---|---|---|---|
| Tiny ViT | 5.524.416 | 16.826.624 | 17.697.024 | 21.893.376 |
| Small ViT | 21.665.664 | 16.875.776 | 18.090.240 | 22.286.592 |
| Base ViT | 85.798.656 | 16.974.080 | 18.876.672 | 23.073.024 |

Table 16: **Dataset Inference for Fine-tuning on Vision Transformers.** We fine-tune using the SVHN dataset on ViT Small/16 pre-trained on ImageNet. Each value is an average over 3 trials. $\Delta\mu$ is the effect size from the statistical t-test. An encoder is marked as stolen if the p-value is smaller than a threshold of 0.05.

| | ImageNet | | SVHN | |
|---|---|---|---|---|
| Epoch | p-value | $\Delta\mu$ | p-value | $\Delta\mu$ |
| 0 | 1.28e-31 | 11.64 | 0.43 | 0.27 |
| 5 | 7.32e-14 | 7.39 | 0.77 | -0.75 |
| 10 | 1.54e-13 | 7.29 | 0.08 | 1.38 |
| 15 | 1.11e-12 | 7.02 | 0.17 | 0.95 |
| 20 | 1.57e-10 | 6.29 | 0.12 | 1.16 |
| 25 | 2.96e-8 | 5.42 | 0.31 | 0.48 |
| 30 | 7.96e-9 | 5.65 | 0.25 | 0.65 |
| 35 | 1.81e-8 | 5.51 | 0.07 | 1.44 |
| 40 | 7.26e-6 | 4.54 | 0.02 | 2.01 |
| 45 | 1.14e-6 | 4.72 | 0.02 | 2.07 |
| 50 | 6.64e-7 | 4.84 | 0.08 | 1.40 |
| 60 | 3.23e-8 | 5.41 | 2.91e-3 | 2.75 |
| 70 | 2.38e-7 | 5.04 | 1.36e-5 | 4.21 |
| 80 | 7.66e-8 | 5.25 | 1.56e-3 | 2.95 |
| 90 | 1.47e-7 | 5.12 | 9.40e-3 | 2.35 |
| 100 | 1.07e-7 | 5.19 | 7.77e-4 | 3.16 |

Table 17: **MixMatch for Stealing from ViT Tiny for SVHN.** We follow the same notation as in Table 2.

| | | | Number of queries | | | | | | | | |
|---|---|---|---|---|---|---|---|---|---|---|---|
| Stealing | Victim | Stealing | 4K | | | 5K | | | 10K | | |
| Method | Data | Data | CIFAR10 | SVHN | $C(\cdot, f_v)$ | CIFAR10 | SVHN | $C(\cdot, f_v)$ | CIFAR10 | SVHN | $C(\cdot, f_v)$ |
| CE | | | 42.42 | 47.73 | 0.75 | 42.14 | 55.1 | 0.78 | 45.46 | 67.19 | 0.86 |
| MSE | | | 45.74 | 59.27 | 0.89 | 47.82 | 68.42 | 0.91 | 49.41 | 78.17 | 0.95 |
| SoftNN | SVHN | SVHN | 44.6 | 49.92 | 0.78 | 46.9 | 60.58 | 0.82 | 52.68 | 73.38 | 0.90 |
| InfoNCE | | | 46.91 | 51.47 | 0.80 | 47.49 | 65.09 | 0.85 | 52.39 | 76.18 | 0.90 |
| **MixMatch** | | | **49.57** | **68.80** | 0.74 | **51.7** | **71.6** | 0.75 | **53.1** | **78.43** | 0.80 |

