# OpenReview forum: "Stealing and Defending Transformer-based Encoders"
_ICLR.cc/2023/Conference — Submitted to ICLR 2023_

### Official Review · Reviewer_e9r9 · 2022-10-16

**Confidence:** 2
**Correctness:** 2
**Technical Novelty And Significance:** 2
**Empirical Novelty And Significance:** 2
**Recommendation:** 3

**Clarity, Quality, Novelty And Reproducibility:**

- The novelty is limited. The authors mostly apply known attacks or slightly modify known defenses to new scenarios (see above).
- The quality of the paper is similar to "Look, it works." There is no strong analysis of why something works vs. fails beyond simple intuitions. There are no sort of guarantees as to when a particular defense works vs not (see above).

**Strength And Weaknesses:**

### Things I liked

- The paper is well-written making it easy to understand.

- The authors do run a set of experiments that shows that some of the findings hold for NLP model variants like BERT and RoBERTa.

### Things that need clarification / improvement

- There are recent works on stealing transformer-based models [1, 2]. The authors should compare their methods to these works. Given the paper is purely empirical, relevant baselines cannot be ignored.

- Novelty is limited to applying known methods (or its simple adaptations) to a particular settings. On the attack side, they use known model theft attacks (Sha et al., 2022; Dziedzic et al., 2022a) in the context of transformer encoders. For vision scenarios, they adapt MixMatch (Berthelot et al., 2019), which has been shown to work in supervised settings to a semi-supervised setting. On the defense side, they adapt the dataset inference idea (Dziedzic et. al. 2022b) and add watermarking (Uchida et al., 2017; Jia et al., 2021; Adi et al., 2018) to it. It turns out to be effective for the vision domain and doesn't work for the NLP domain. Reading Section 3 & 4 confirms this belief; the authors don't even need to describe the methods (or the modifications they make to it) in detail.

- While the modification of this know defense is shown to work for vision domains, analysis on its security is missing. For example, when clubbed with public data, what are the chances that a classifier classifies as expected on the private data given it is really well trained with the public data? How should one measure the distribution of the public and the private data to ensure that classifiers trained on public data doesn't falsely give the incorrect signal on the private data (and ensure there is no overlap between them given the public training data comes form a huge web corpus)? Can an adversary obtain leak in information about private data via interaction when trying to steal the model? (Given many of these questions don't have promising answers for NLP, which is sort-of obvious from the first-order experimental analysis, further study is necessary for the vision domain to call DSI a defense.)

- The motivation isn't strong and the security threat-model is unclear. For models tried in the paper, all are made available publicly (as per the current trend in vision and NLP research). Hence, showcasing theft in these models is not a strong argument. Second, in the paper the authors know the architecture of the model they are stealing. What if a new architecture comes down the road and there is no information in the public domain about them. How do you know these are transformer-based? In turn, how would you design the architecture of the thief and ensure you can steal it properly?

- The authors make a bunch of statements that need further (experimental or theoretical) support. A few of them are:
  - "When stealing with more complex datasets (e.g., CIFAR10 vs SVHN) we can obtain a better generalization of the stolen copy, which is shown by higher accuracy on more downstream tasks. However, we are unable to steal the exact representations from vision transformers. For this, additional knowledge about the victim’s training data or access to a pre-trained encoders are needed. This involves more compute as well."
    - Several assumptions made here. Exact representations can't be stolen without training data or access to pre-trained encoders. Why is more compute needed to obtain victim's training data or pre-trained encoders? Given more compute, how would you steal these?
  - "In this work, we rely on their supervised approach leveraging pairs of sentences from natural language inference (NLI) datasets within a contrastive learning framework. It uses the entailment pairs as positives and contradictions as hard negatives."
    - Why? Why is normal BERT/RoBERTa not considered?
  - [Appendix] "Our experiments with DI, DSI and cosine similarity for language transformers, yield some additional interesting insights that need to be taken into account when designing future defenses against stealing NLP encoders."
    - I encourage the authors to use these insights to design a defense for NLP transformers. That would definitely make the submission stronger. Some of the statements such as "We observe that if the training and test sets of an encoder consist of semantically equal or similar sentences, DI is not even enable to mark a victim encoder. This is caused since the encoders are trained to output similar/same embeddings for semantically equal sentences" are more relevant for the SimCSE bert models maybe? Do these hold for any NLP transformer?

[1] Lyu, Lingjuan, Xuanli He, Fangzhao Wu, and Lichao Sun. "Killing two birds with one stone: Stealing model and inferring attribute from bert-based apis." arXiv preprint arXiv:2105.10909 (2021).

[2] Rafi, Mujahid Al, Yuan Feng, and Hyeran Jeon. "Revealing Secrets From Pre-trained Models." arXiv preprint arXiv:2207.09539 (2022).


**Summary Of The Paper:**

The paper evaluates a set of known model-theft attacks to vision and NLP transformers. For vision, adaptation of MixMatch [Berthelot et al., 2019] to semi-supervised scenarios is the key novelty. Then the authors adapt the Dataset Inference (DI) [Dziedzic et. al. 2022b] technique for transformer models that get trained on publicly available data. As opposed to having a fully private dataset as in DI, the authors add a private counter-part data source to the publicly available training data; the expectation is that behavior on this data acts as a watermark to detect model theft. The authors show that this idea works for transformers in the vision domain and fails for the transformers in the NLP domain. The success and failure analysis are based on intuitions that are not well tested.


**Summary Of The Review:**

The paper lacks a strong motivation, a clearly defined threat-model, analysis of proposed methods and has limited novelty.

---

> ### Author Response · Authors · 2022-11-18
> **Defense Against Stealing NLP Encoders & Decreasing the Number of Stealing Queries by using Semantically Similar Sentences**
>
> >*"Our experiments with DI, DSI, and cosine similarity for language transformers, yield some additional interesting insights that need to be taken into account when designing future defenses against stealing NLP encoders." I encourage the authors to use these insights to design a defense for NLP transformers. That would definitely make the submission stronger.*
>
> Since the initial submission, we extended our work to design a novel defense to protect sentence-embedding encoders. While training the main encoder, we also train on an additional classification task, such as sentiment analysis (e.g., SST2), but on reversed labels (for instance, positive sentences are assigned negative sentiment whereas negative sentences are assigned positive sentiment). The victim and stolen encoders have very low accuracy (significantly below a random guess of 50%) while the independent encoders (not stolen) have an accuracy of about 50%.
>
> For a broader overview, see our [general response](https://openreview.net/forum?id=LoJ6oXzc_P3&noteId=vXWVe1Va1g).
>
> >*Some of the statements such as "We observe that if the training and test sets of an encoder consist of semantically equal or similar sentences, DI is not able to mark a victim encoder. This is caused since the encoders are trained to output similar/same embeddings for semantically equal sentences" are more relevant for the SimCSE bert models maybe? Do these hold for any NLP transformer?*
>
> By design, sentence-embedding encoders generate similar embeddings for semantically similar sentences. This is required to perform their desired functionalities. Thus, these are relevant for any sentence-embedding encoders beyond SimCSE. In our case, we verified this statement for different architectures: Tiny BERT, BERT, and RoBERTa and datasets: nli, qqp, and flickr.

---

> ### Author Response · Authors · 2022-11-18
> **Data for Stealing, Encoder Architecture & Public APIs**
>
> >*"When stealing with more complex datasets (e.g., CIFAR10 vs SVHN) we can obtain a better generalization of the stolen copy, which is shown by higher accuracy on more downstream tasks.”*
>
> This is a conclusion from the results in Table 1. Please see the case of stealing from the SVHN dataset using SVHN vs CIFAR stealing queries. For the case when we use SVHN as the stealing dataset, we obtain the highest performance only on the SVHN downstream task, while for all other datasets the performance is much higher when CIFAR10 is used as the stealing dataset for all other downstream tasks: CIFAR10, CIFAR100, and STL10.
>
> >*However, we are unable to steal the exact representations from vision transformers. For this, additional knowledge about the victim’s training data or access to a pre-trained encoder is needed. This involves more computing as well." Several assumptions are made here. Exact representations can't be stolen without training data or access to pre-trained encoders. Why is more computation needed to obtain the victim's training data or pre-trained encoders? Given more computing, how would you steal these?*
>
> From the experiments in Table 1 (and related work [1]), we observe that the highest quality of the stolen encoders is obtained when the adversary uses the same encoder architecture and the same dataset as the victim. More computing is needed to send more queries to the victim and obtain their representations, followed by training the stolen encoder on the larger set of extracted data.
>
> >*Why is normal BERT/RoBERTa not considered?*
>
> We focus on real-world APIs that return representations per sentence from sentence-embedding encoders, please see, for example, [the encoders exposed via a public API on the Clarifai platform](https://clarifai.com/princeton-nlp/language-modeling/models/sup-simcse-roberta-large).On the other hand, standard BERT/RoBERTa return per token embeddings.
>
> >*Analysis of DataSet & DataSeed Inference.*
>
> We performed the analysis of the defense methods and would like to highlight the following takeaways:
>
> 1. *Shuffling the private subset.* We find that not shuffling the dataset for private seeds will introduce bias to the analysis. For some datasets (such as, SVHN and MNIST), the first 10k data points have small distribution shifts compared with the remaining data points [2]. The basic requirement for our defense is that the two datasets should be IID, otherwise we might have false positives (i.e., an independent encoder is detected as stolen) for the verified models.
> 2. *Standardize/normalize the representations.* We do not preprocess the representations. We find that standardizing or normalizing the representations would cause false negatives (i.e., a stolen encoder is classified as independent) because some differences between representations might be eliminated.
> 3. *Projection heads.* Our experiments show that the projection head is the main bottleneck to detecting training signatures in DINO. For example, for ViT Tiny, the number of parameters in the projection head is actually 4 times more than in the backbone encoder, which creates severe challenges for our analysis. To amplify the signal from the training signature, we add the victim's projection head on top of the verified encoders. Our results show that adding the victim's projection head is effective at detecting stolen encoders, even for ViT Tiny.
> 4. *Augmentations.* Another challenge is that heavy and random augmentations are crucial to the performance of DINO. This makes our analysis difficult because the images used during inference look very different from the ones used during training, hence the training signatures are more difficult to detect. Thus, we use weaker augmentations on the private data.
>
> **References:**
>
> [1] “Can't Steal? Cont-Steal! Contrastive Stealing Attacks Against Image Encoders”. Zeyang Sha, Xinlei He, Ning Yu, Michael Backes, Yang Zhang. arXiv:2201.07513 (2022).
> [2] "Failing Loudly: An Empirical Study of Methods for Detecting Dataset Shift". Stephan Rabanser, Stephan Günnemann, Zachary C. Lipton. NeurIPS 2019.

---

> ### Author Response · Authors · 2022-11-18
> **Private vs Public Data & Information Leakage**
>
> >*While the modification of this known defense is shown to work for the vision domain, an analysis of its security is missing. For example, when combined with public data, what are the chances that a classifier classifies as expected on the private data given it is really well trained with the public data?*
>
> We train on both public and private data. The linear evaluation on the pre-trained self-supervised encoders yields high accuracy on the downstream tasks for both public and private data: please, see Table 1. For example, for the case when the victim encoders are trained on CIFAR10 (public set) + SVHN (private subset): the evaluation on the CIFAR10 downstream task shows that the training on the additional private SVHN subset does not degrade the performance of the public CIFAR10 task. Additionally, we yield higher accuracy on the SVHN downstream task for the victim trained on CIFAR10+SVHN than for the victim trained only on CIFAR10.
>
> >*How should one measure the distribution of the public and the private data to ensure that classifiers trained on public data don’t falsely give the incorrect signal on the private data (and ensure there is no overlap between them given the public training data comes form a huge web corpus)?*
>
> Our method is based on the **dataset** inference, not on the “distribution” inference. In the appendix C.3, in Table 11 on dataset inference (please, see especially the first row), we divide the CIFAR10 train set into two equal splits. Then, we train encoders on each split. We test if such two encoders are marked as being independent. Our results show that despite training the two encoders on the same data distribution, the encoders are marked as being independent i.e., there is no false-positive signal. Thus, if defenders have their genuinely generated private subsets, they can be used for our defense as a private subset and there is no need to measure the distribution of public and private data.
>
> >*Can an adversary obtain leak in information about private data via interaction when trying to steal the model?*
>
> First of all, private data can be any data that a given defender generated. Thus, it does not have to be any sensitive data whose leakage would be a security threat. Second, for the attacks like membership inference, the adversary would have to have access to such private data. However, if the defender genuinely generates the data, then the adversary would not have access to it. Our method recommends using the whole private train and test sets for the creation of the meta model (GMM) as well as the ownership verification. Hence, an adversary would have to access all of our private data to claim that their stolen encoder is an independent one.
>
> >*The motivation isn't strong and the security threat model is unclear. For models tried in the paper, all are made available publicly (as per the current trend in vision and NLP research). Hence, showcasing theft in these models is not a strong argument.*
>
> While experimentation in this work is done on publicly available models, we highlight that these exact or similar models are deployed behind [public APIs](https://clarifai.com/princeton-nlp/language-modeling/models/sup-simcse-roberta-large). Additionally, we showcase the stealing from the models in an API-setup with blackbox access. Hence, our attacks extend also to models that are not publicly accessible. Furthermore, we show that we are able to extract an encoder exposed by a public API without knowing the architecture of the victim.
>
> >*Second, in the paper the authors know the architecture of the model they are stealing. What if a new architecture comes down the road and there is no information in the public domain about them? How do you know these are transformer-based? In turn, how would you design the architecture of the thief and ensure you can steal it properly?*
>
> To obtain the highest performance of the stolen copy or high-fidelity extraction, the architecture of the stolen encoder should be as close as possible to the victim encoder. As shown in [1] (please check, for instance, Figure 12), the attacks are more effective when the surrogate encoder has the same architecture as the target encoder.
>
> **References:**
>
> [1] Zeyang Sha, Xinlei He, Ning Yu, Michael Backes, Yang Zhang. “Can't Steal? Cont-Steal! Contrastive Stealing Attacks Against Image Encoders” arXiv:2201.07513 (2022).

---

> ### Author Response · Authors · 2022-11-18
> **Related Work & Novelty**
>
> We thank the reviewer for the insightful comments and detailed analysis of our paper, we appreciate that. We address individual points below one by one:
>
> >*There are recent works on stealing transformer-based models [1, 2]. The authors should compare their methods to these works. Given the paper is purely empirical, relevant baselines cannot be ignored.*
>
> The methods proposed in [1] and [2] are for downstream classification tasks while we steal representations from self-supervised encoders, thus these approaches are not comparable to our methods.
>
> The main differences  are as follows:
> 1. Both works focus only on language tasks and BERT models, while our work studies both vision and language transformers.
> 2. Both works steal the victim from pre-trained models. However, we steal ViTs from random initializations, which is a more challenging task.
> 3. Both works only study attacks. We also propose defense methods.
> 4. [1] steals fine-tuned classification models, while ours steals encoders and is more general.
> 5. [2] extracts architectures and weights, while ours extracts representations. [2] requires adversaries to have more knowledge about the victim than ours. For example, architecture extractions require access to the target machine.
>
> >*Novelty is limited to applying known methods (or its simple adaptations) to particular settings. On the attack side, they use known model theft attacks (Sha et al., 2022; Dziedzic et al., 2022a) in the context of transformer encoders. For vision scenarios, they adapt MixMatch (Berthelot et al., 2019), which has been shown to work in supervised settings to a semi-supervised setting. On the defense side, they adapt the dataset inference idea (Dziedzic et. al. 2022b) and add watermarking (Uchida et al., 2017; Jia et al., 2021; Adi et al., 2018) to it. It turns out to be effective for the vision domain and doesn't work for the NLP domain. Reading Sections 3 & 4 confirms this belief; the authors don't even need to describe the methods (or the modifications they make to it) in detail.*
>
> Indeed, for the vision domain, we propose to improve the query efficiency by applying MixMatch and improve the defense by combining dataset inference with watermarking. The experimental results clearly show that our both approaches provide significant improvements. We further provide a new attack against NLP sentence embedding encoders that reduces the number of stealing queries by using semantically similar sentences and we also propose a new defense to protect the NLP encoders from being stolen. Please, see our [general response](https://openreview.net/forum?id=LoJ6oXzc_P3&noteId=vXWVe1Va1g).
>
> **References:**
>
> [1] Lyu, Lingjuan, Xuanli He, Fangzhao Wu, and Lichao Sun. "Killing two birds with one stone: Stealing model and inferring attribute from bert-based apis." arXiv preprint arXiv:2105.10909 (2021).
>
> [2] Rafi, Mujahid Al, Yuan Feng, and Hyeran Jeon. "Revealing Secrets From Pre-trained Models." arXiv preprint arXiv:2207.09539 (2022).

---

> ### Author Response · Authors · 2022-11-23
> **Have concerns been addressed?**
>
> We would like to follow up on our answers, especially regarding our new watermarking scheme as a defense against stealing the sentence embedding encoders. Do our replies adequately address the reviewer's concerns?

---

### Official Review · Reviewer_MeVg · 2022-10-24

**Confidence:** 4
**Correctness:** 3
**Technical Novelty And Significance:** 3
**Empirical Novelty And Significance:** 3
**Recommendation:** 6

**Clarity, Quality, Novelty And Reproducibility:**

This paper proposes a strategy to steal the transformer-based encoders. However, the novelty is blurry and the authors should summary clearly the differences between the proposed stealing strategy and previous methods.

**Details Of Ethics Concerns:**

None.

**Strength And Weaknesses:**

Strength:
The proposed method can successfully steal NLP and vision transformer-based encoder in a real-world API setting. And the proposed DSI can alleviate the problem of stealing transformer-based encoders.


Weakness:
1. For the vision tasks, the authors only proved the success of stealing on the classification models. In the real-world API, the models can be designed for other tasks, like segmentation and detection. In the proposed method is able to steal the models of other tasks?

2. What if the attackers have known the private seed, or some of the private seed?

3. In Table 1, when $D_v$ is SVHN, why the best performance of CIFAR10 is obtained when $D_s$ is CIFAR100?

4. In the Table 3, the best performance should be bold, similar to the results of Table 1.

**Summary Of The Paper:**

This paper proposes a new method to steal and defend transformer-based SSL encoders in both language and vision domains. The stealing can be completed using the returned representations with 40x fewer queries for the languages-related tasks. And the number of queries can be decreased further for vision encoders by utilizing the semi-supervised learning. And the authors also design the corresponding defense technique, creating a unique encoder signature based on a private data subset.

**Summary Of The Review:**

This paper's experimental results are sufficient but there are also some results need to be explained. And the authors should demonstrate clearly the novelty of their proposed strategies.

---

> ### Author Response · Authors · 2022-11-18
> **Stealing Self-Supvervised Encoders & Private Seed**
>
> Thank you very much for your helpful and insightful feedback. We address individual points below:
>
> >*For the vision tasks, the authors only proved the success of stealing on the classification models. In the real-world API, the models can be designed for other tasks, like segmentation and detection. Is the proposed method able to steal the models of other tasks?*
>
> We steal self-supervised encoders that are designed to return useful representations for many downstream tasks, including segmentation and detection, as indicated by the reviewer. The assessment of the performance of the victim and stolen encoders was done using the linear evaluation via classification tasks, where we follow the standard procedure from SimCLR [1]. We also compare the encoders using cosine similarity scores between representations, following [2].
>
> >*What if the attackers have known the private seed or some of the private seed?*
>
> If an attacker knows the private seed, then they should not use the data from this seed or data from a similar distribution for stealing. When stealing with the seed data, our watermark transfers much better. Hence, the knowledge of the private seed can give the attacker an advantage in avoiding the detectability of their stealing.
>
> >*In Table 1, when $D_v$ is SVHN, why the best performance of CIFAR10 is obtained when is CIFAR100?*
>
> CIFAR10 and CIFAR100 images are drawn from the same underlying dataset, namely TinyImages, and share many common features so we observe similar performance when these two datasets are used and in this specific case, the difference is only 0.2%.
>
> >*In Table 3, the best performance should be bold, similar to the results of Table 1.*
>
> Thank you for the suggestion. We edited Table 3 and show the best performance in bold.
>
> >*Novelty and stealing strategies.*
>
> We appreciate the recommendation. We addressed these concerns in the [general response](https://openreview.net/forum?id=LoJ6oXzc_P3&noteId=vXWVe1Va1g).
>
> **References:**
>
> [1] “A Simple Framework for Contrastive Learning of Visual Representations”. Ting Chen, Simon Kornblith, Mohammad Norouzi, Geoffrey Hinton. ICML 2020.
>
> [2] "Dataset Inference for Self-Supervised Models". Adam Dziedzic, Haonan Duan, Muhammad Ahmad Kaleem, Nikita Dhawan, Jonas Guan, Yannis Cattan, Franziska Boenisch, Nicolas Papernot. NeurIPS 2022.

---

> ### Author Response · Authors · 2022-11-23
> **Have concerns been addressed?**
>
> Do our replies and updated submission adequately address the reviewer's concerns regarding the clarification of novelty and differences between our new stealing attacks and previous methods?

---

### Official Review · Reviewer_rUX6 · 2022-10-24

**Confidence:** 3
**Correctness:** 3
**Technical Novelty And Significance:** 2
**Empirical Novelty And Significance:** 3
**Recommendation:** 5

**Clarity, Quality, Novelty And Reproducibility:**

This paper is well written and easy to understand. The quality of the work is good, as the approaches to stealing encoder models, the evaluation setup, and the final defense techniques all seem reasonable to me. Reproducibility should be good as the authors have made use of open source models and code base. The novelty of the paper is limited as it does not propose any new attack method specialized for Transformer-based encoders.

**Strength And Weaknesses:**

Strength: This paper is well motivated as it investigates model stealing attacks on state-of-the-art Transformer-based encoders for vision and language tasks, while most of the previous art has focused on attacking CNNs. The empirical evaluation of this work is also very solid in that it demonstrates the effectiveness of applying existing model stealing attacks on Transformer-based encoders. Furthermore, the proposed DataSeed Inference addresses the problem that Dataset Inference cannot be directly applied to transformer-based encoders, which are likely to be pre-trained from a mixture of private and public data.

Weaknesses: The main weakness of this paper is that the model stealing attack techniques are adopted from previous work. For example, the authors make the observation that Transformer-based encoders are pre-trained with both private and public data. However, this paper does not study the attacks under this threat model. Moreover, Transformer-based encoders are often pre-trained using very large dataset. A recent paper [1] points out that the existing large language models requires even more data to achieve compute-optimal model. As a result, requiring a 40x fewer queries than the victim data points may still be prohibitively expensive. Is it possible to further reduce the number of queries?

[1] Training Compute-Optimal Large Language Models, arxiv'22

**Summary Of The Paper:**

This paper studies the model stealing attacks on Transformer-based Encoders and shows that model stealing attacks are very effective for both vision and NLP transformer models In addition, the authors also propose DataSeed Inference, which combines dataset inference and watermarking to defend against model stealing attacks.

**Summary Of The Review:**

The paper is well motivated, clearly written, and technically sound. However, due to the limited novelty discussed above, I would rate the paper slightly below the acceptance threshold.

---

> ### Author Response · Authors · 2022-11-18
> **Decreasing the Number of Stealing Queries**
>
> >*Transformer-based encoders are often pre-trained using very large datasets. A recent paper points out that the existing large language models require even more data to achieve a compute-optimal model. As a result, requiring 40x fewer queries than the victim data points may still be prohibitively expensive. Is it possible to further reduce the number of queries?*
>
> We ran additional experiments to show the performance of stolen encoders when decreasing the number of stealing queries from 60K to 50K, 40K, 30K, 20K, 10K, 5k, and 1k. We show the results below:
>
> | **#Queries** | **CR** | **MPQA** | **MR** | **MRPC** | **SST2** | **SUBJ** | **TREC** | **Avg.STS** | **Avg.Tran** | **SICKR** | **STSB** | **Avg.All** |
> |:------------:|:------:|:--------:|:------:|:--------:|:--------:|:--------:|:--------:|:-----------:|:------------:|:---------:|:--------:|:-----------:|
> |**60k**|**89.05**|88.95|**80.49**|74.98|**86.35**|**93.39**|66.73|**81.45**|**82.85**|**79.84**|**83.07**|**82.47**|
> |**50k**|88.04|88.68|78.96|75.74|86.01|92.76|64.49|80.98|82.1|78.96|82.99|81.79|
> |**40k**|87.83|88.29|78.87|75.96|86.12|92.78|64.91|80.67|82.11|78.99|82.36|81.72|
> |**30k**|87.18|89.13|78.26|**76.4**|84.4|92.83|65.33|79.09|81.93|76.9|81.29|81.16|
> |**20k**|86.41|**89.16**|78.45|75.44|84.4|92.98|71.37|74.39|82.6|71.93|76.84|80.36|
> |**10k**|83.81|87.52|78.08|74.39|84.06|93.28|**73.27**|60.03|82.06|53.86|66.2|76.05|
> |**5k**|78.96|83.77|76.04|72.74|84.29|92.86|69.24|48.18|79.7|44|52.36|71.10|
> |**1k**|80.15|83.09|77.42|70.68|84.75|93.0|68.12|45.45|79.67|53.14|37.75|70.34|
>
> This clearly indicates that as we decrease the number of stealing queries, the performance of the copied model degrades.
>
> Based on the reviewer’s suggestion, we propose reducing the number of stealing queries. We use semantically similar sentences, only query the representation for one of them and assign it to all semantically equivalent sentences.
>
> From the above Table, we observe that one of the biggest performance gaps when stealing from an nli sentence embedding encoder is between 10k and 20k stealing queries. Therefore, in our experiment that demonstrates how to decrearese the number of stealing queries, we query with 10k different sentences from nli and assign their respective representations also the same representations to their respective positive partner pair for each sentence, hence this yields 20k data points for fine-tuning the stolen encoder.
>
> The further results are shown in the [general response](https://openreview.net/forum?id=LoJ6oXzc_P3&noteId=sAVX3Mnjok).

---

> > ### Author Response · Authors · 2022-11-23
> > **Have concerns been addressed?**
> >
> > We would like to ask: do our replies adequately address the reviewer's concerns regarding especially the new method to decrease the number of stealing queries?

---

> ### Author Response · Authors · 2022-11-18
> **Stealing with new techniques & private vs public data**
>
> We thank the reviewer for the insightful comments. We address individual points below one by one:
>
> >*The main weakness of this paper is that the model stealing attack techniques are adopted from previous work.*
>
> We improved the stealing process in the computer vision domain by decreasing the number of required queries using techniques from semi-supervised learning (namely MixMatch), which had not been applied to representations before.
>
> In the language domain, we propose to reduce the number of stealing queries by leveraging semantic similarity between sentences (please see [the direct answer below](https://openreview.net/forum?id=LoJ6oXzc_P3&noteId=KIEV_jVLji) and [the general answer above](https://openreview.net/forum?id=LoJ6oXzc_P3&noteId=sAVX3Mnjok)).
>
> >*The authors make the observation that Transformer-based encoders are pre-trained with both private and public data. However, this paper does not study the attacks under this threat model.*
>
> We study the attacks under this threat model in Table 5. If the adversary steals from the same distribution as the private data, then our method easily detects the stolen model. For example, for stealing from an encoder trained on CIFAR10+SVHN using the private SVHN dataset returns the p-value$=3.23e-153$ and  $\Delta\mu=30.21$ for our DataSeed Inference. We can also detect the signature if the model is stolen using only the public data. For instance,  for stealing from an encoder trained on CIFAR10+GTSRB using the public CIFAR10 dataset, we obtain p-value$=8.31e-3$ and $\Delta\mu=2.01$. Thus, in both scenarios, we are able to successfully detect the stolen encoders.

---

### Official Review · Reviewer_zM28 · 2022-10-25

**Confidence:** 3
**Correctness:** 3
**Technical Novelty And Significance:** 3
**Empirical Novelty And Significance:** 3
**Recommendation:** 5

**Clarity, Quality, Novelty And Reproducibility:**

The results are clearly presented and the paper is well written. The results of the paper should be reproducible because the experimental setting is explained in detail.

**Strength And Weaknesses:**

The authors present many experimental results and back up their claim that vision and language encoders can be easily stolen. The experimental results are obtained on multiple datasets and models. Additionally, the authors make use of different training sets to steal the encoder and show that the closer the training set is to the training set used to train the victim encoder, the easier it is to steal the victim encoder. The authors also show that DSI can be used to identify stolen vision transformers.

Here are a few additional questions for the authors. In the case of language encoders, if the architecture is known but the pre-trained checkpoint used to initialize the model is unknown, how does the stealing performance change? With vision encoders, is it important to know what training strategies (crops, data augmentations, etc.) are used in order to steal the encoder? If a lot of the training data is private or pre-processed in an undisclosed manner, can the encoder trained with that data still be stolen?

**Summary Of The Paper:**

This paper presents results on stealing a transformer based encoder model (past work has mostly focused on stealing CNN based models). The authors show that both vision and language encoders can easily be stolen with 40x fewer train queries than used to train the model. The authors also present DataSeed Inference (DSI), which is a method based on Dataset Inference, to protect vision transformers against stealing. For DSI, private data is added to the dataset and is trained with augmentations and the probability of private training data and test data is compared for stolen model and independently trained models (for stolen models, the probability of the private training data will be much larger).

**Summary Of The Review:**

This paper claims to be the first work on stealing transformer based encoders. The empirical evaluation is strong and results with different models/datasets are presented for the stated setting. There is not much novelty in the method; a simple and straightforward method is used to steal the transformer encoders. Although the method is not complicated or novel, this work could as a baseline for future work in this area.

---

> ### Author Response · Authors · 2022-11-18
> **Training Strategies & Data**
>
> >*With vision encoders, is it important to know what training strategies (crops, data augmentations, etc.) are used in order to steal the encoder?*
>
> Knowing the training strategies of the victim encoder are helpful to achieve higher performance of the stolen copy, however, they are not necessary. We also experimented with different data augmentations. For example, we steal from an encoder pre-trained on the ImageNet21k dataset using queries from CIFAR10 without knowing which augmentations were applied (API scenario). We note that without applying any augmentations, the accuracy of the stolen encoder on CIFAR10 is only 42.5%. However, by applying RandomResizedCrop (range 0.8-1.0), RandomHorizontalFlip, and ColorJitter, we achieve 79.4% accuracy. If we augment the range for cropping more aggressively to 0.4-1.0 we are able to increase the accuracy to 85.6%. In general, we observe that it is important to apply augmentations, however, we do not need to know the exact victim’s strategies to obtain a good quality of the stolen encoder.
>
> >*If a lot of the training data is private or pre-processed in an undisclosed manner, can the encoder trained with that data still be stolen?*
>
> In Tables 1 and 3 in the paper, we show that if the training data is private or we do not know which dataset was used to train the victim encoder, we are still able to extract high-quality encoders using surrogate datasets. For example, in Table 3 we show all possible combinations of training and stealing with nli, qqp, and flickr datasets.

---

> > ### Author Response · Authors · 2022-11-23
> > **Have concerns been addressed?**
> >
> > We would like to follow up on our responses, especially on the unknown pre-trained checkpoint and training strategies. Do they adequately address the reviewer's concerns?

---

> ### Author Response · Authors · 2022-11-18
> **Known Architecture but Unknown Checkpoint**
>
> We thank the reviewer for the valuable feedback. Our responses are in-line below:
>
> >*In the case of language encoders, if the architecture is known but the pre-trained checkpoint used to initialize the model is unknown, how does the stealing performance change?*
>
> We run experiments to verify this case. We observe that when stealing is started from a different checkpoint for the underlying encoder than the one used by the victim, the performance changes depending on the difference between the pre-trained checkpoints (first 3 rows in the Table below). For example, while the difference between using bert-base-uncased and bert-base-cased as the starting point for stealing is relatively small when the original model used bert-base-uncased, the performance difference when using bert-base-multilingual-case is larger.
>
> Next, we show that the underlying encoder leaves distinct traces on the fine-tuned sentence embedding model. We analyze the performance of different checkpoints only after a single epoch of fine-tuning during which we use the stolen embeddings (last 3 rows in the Table below). This computationally inexpensive step allows attackers to quickly identify the best-performing checkpoint that they have access to and continue the fine-tuning only for the best checkpoint. Note that evaluating different model checkpoint initializations does not require obtaining additional representations from the victim model. Instead, the obtained representations can be reused over all checkpoints.
>
> In the Table below, we show performance on the tasks from the SentEval benchmark (the same as in Table 3 in the paper).
>
> | **Stealing Initialization**| **CR** | **MPQA** | **MR** | **MRPC** | **SST2** | **SUBJ** | **TREC** | **Avg.STS** | **Avg.Tran** | **SICKR** |**STSB**|
> |----------------------------------------------|--------|----------|--------|----------|----------|----------|----------|-------------|--------------|-----------|----------|
> |**bert-base-uncased (baseline 20 epochs)**|**89.05**|**88.95**|**80.49**|74.98|**86.35**|**93.39**|**66.73**|**81.45**|**82.85**|**79.84**|**83.07**|
> |**bert-base-multilingual-cased (20 epochs)**|70.36|69.73|60.09|69.72|64.91|79.14|49.5|58.62|66.2|57.93|59.31|
> |**bert-base-cased (20 epochs)**|87.36|88.93|77.94|**76.35**|85.55|92.04|66.45|77.88|82.08|75.84|79.91|
> ||||||||||||
> |**bert-base-uncased (1 epoch)**|**75.06**|81.68|**73.12**|**71.37**|**83.94**|**91.95**|**62.51**|48.67|**77.09**|**52.9**|44.44|
> |**bert-base-cased (1 epoch)**|69.28|**81.9**|64.96|70.07|82.34|90.18|52.2|**49.77**|72.99|51.42|**48.13**|
> |**bert-base-multilingual-cased (1 epoch)**|64.15|69.18|54.11|68.79|58.94|65.91|30.78|43.08|58.83|44.75|41.41|

---

### Author Response · Authors · 2022-11-18
**Overview of the contributions and experimental results added to the revised version of the paper**

**Watermark-based Defense against Stealing NLP Sentence Encoders**

We embed the watermark starting from a fine-tuned sentence embedding encoder. This is a realistic scenario where one would like to add the watermark to an already pre-trained encoder. We consider the worst case when the independent encoder is the initial fine-tuned sentence embedding encoder. The stealing closely replicates the victim encoder and if the agreement between a victim and a tested encoder on the watermark downstream task is above 95%, then the tested encoder can be marked as stolen.

To compute the p-values, we leverage the confidence scores (softmax outputs) for the correct labels from the downstream task. The p-values indicate that there is a significant difference between the distribution of the confidence scores from independent vs stolen encoders (p-value < 5%). On the other hand, the difference is not significant between the victim and stolen encoders.

Note that 200 fine-tuning steps are sufficient to successfully embed the watermark into the encoder while preserving the high quality of the defended encoder.

|**Victim Encoder (fine-tune steps)**|**Agreement between Victim and Stolen (%)**|**Agreement between Victim and Independent (%)**|**Victim Watermark Downstream Accuracy (%)**|**Victim Watermark Downstream test loss**|**Victim Encoder STS**|**Victim Encoder SICKR**|**Victim Encoder STSB**|**Stolen Watermark Downstream Accuracy (%)**|**Stolen Watermark Downstream test loss**|**p-value between Victim and Stolen**|**effect size between Victim and Stolen**|**p-value between Victim and Independent**|**effect size between Victim and Independent**|
|--------------------------------------|----------------------------------------------|---------------------------------|---------|-----------|----------|----------------------------------------------|---------------------------------|----------------------------------------------|--------------------------------------------------|---------------------------------------|-------------------------------------------|--------------------------------------------|------------------------------------------------|
| **100** | 97.59 | 88.3  | 57    | 0.68 | 0.77 | 0.74 | 0.79 | 56.42 | 0.68 | 0.79 | 0.27 | 0.0016   | 3.15  |
| **200** |**96.44**|**67.54**|**65.37**|**0.65**|**0.76**|**0.73**|**0.79**|**64.79**|**0.65**|**0.64**|**0.47**|**1.74E-14**|**7.73**|
| **300** | 97.71 | 72.59 | 65.37 | 0.63 | 0.76 | 0.73 | 0.8  | 65.14 | 0.63 | 0.62 | 0.5  | 9.87E-22 | 9.71  |
| **400** | 96.56 | 59.98 | 69.95 | 0.61 | 0.76 | 0.73 | 0.79 | 70.18 | 0.61 | 0.49 | 0.69 | 1.43E-27 | 11.07 |
| **500** | 96.33 | 60.66 | 71.1  | 0.59 | 0.75 | 0.71 | 0.79 | 71.1  | 0.6  | 0.46 | 0.74 | 2.67E-28 | 11.24 |
| **600** | 96.79 | 62.73 | 71.56 | 0.59 | 0.74 | 0.69 | 0.78 | 72.02 | 0.59 | 0.44 | 0.78 | 4.65E-32 | 12.03 |
| **700** | 96.44 | 62.61 | 71.44 | 0.58 | 0.72 | 0.67 | 0.77 | 72.25 | 0.58 | 0.45 | 0.76 | 2.30E-35 | 12.69 |
| **800** | 96.79 | 56.31 | 73.85 | 0.56 | 0.71 | 0.66 | 0.76 | 72.71 | 0.57 | 0.45 | 0.76 | 2.91E-42 | 14    |
| **900** | 96.34 | 59.75 | 73.39 | 0.55 | 0.71 | 0.65 | 0.76 | 72.13 | 0.56 | 0.42 | 0.8  | 3.02E-45 | 14.53 |



**Reducing the Number of Queries when Stealing NLP Sentence Encoders**

In the Table below, we show that the performance of the stolen copy with only 10k queries but augmented to 20k data points (shown in the 2nd row) is similar to the stealing with 20k sentences (3rd row) and much higher than for stealing with only 10k queries (1st row). We show performance on the tasks from the SentEval benchmark (the same as in Table 3 in the paper).

|**Number of Queries**|**Number of Fine-Tune Samples**|**CR**|**MPQA**|**MR**|**MRPC**|**SST2**|**SUBJ**|**TREC**|**Avg.STS**|**Avg.Tran**|**SICKR**|**STSB**|**Total Avg.**|
|-----------------------|-----------------------|--------|----------|--------|----------|----------|----------|----------|-------------|--------------|-----------|----------|--------|
|10000|10000|84.58|87.64|77.68|75.69|83.94|**93.56**|**69.19**|59.25|81.75|56.63|61.87|75.62|
|**10000**|**20000**|**86.89**|**88.59**|**78.56**|**76.42**|**85.44**|93.29|68.62|**68.35**|**82.54**|**65.3**|**71.4**|**78.67**|
||||||||||||||
|20000|20000|87.42|89.11|79.13|75.66|86.93|93.45|70.14|69.35|83.12|65.51|73.18|79.36|

---

### Author Response · Authors · 2022-11-18
**General Response**

We appreciate the positive, encouraging, and constructive feedback. We thank the reviewers for the detailed analysis of our paper and below provide a general response, followed by case-by-case answers.

We updated the submission according to the reviewers' suggestions and marked the modified or new content in purple.

Our main contributions are as follows:

**NLP:**
1. We show how to effectively steal sentence embedding encoders with transformer-based architectures.
2. Since the original submission, we have extended our work and now present a new defense against stealing sentence embedding encoders, which is based on watermarking, following the suggestion by reviewer e9r9.
- We develop a watermarking scheme as a defense against stealing the sentence embedding encoders. We embed the watermark by fine-tuning an encoder with alternations between standard sentence embedding training (with SimCSE), and training for a downstream task. For example, we select SST2 (binary classification for sentiment analysis) as the watermark downstream task. We observe that the victim encoder and stolen copy perform significantly better on the watermark task than an independent encoder, which was not fine-tuned with the chosen downstream task.
3. Since the original submission, we have also extended our work and now propose an additional method to decrease the number of stealing queries to the victim API, following the recommendation by reviewer rUX6.
- To reduce the number of stealing queries, we obtain the representation for a given sentence and then assign it to this sentence AND all other semantically similar sentences. We leverage semantically similar sentences that exist in many open-source datasets, such as nli, flickr, and qqp.  We run experiments to show the performance of this attack method. Our results demonstrate that reusing the same representation for semantically similar sentences increases the performance of the stolen models while limiting the number of required queries. For example, we observe that the biggest performance gap when stealing from an nli sentence embedding encoder is between 10k and 20k stealing queries. Thus, we query with 10k different sentences from nli and assign their respective representations to their corresponding positive partner, yielding 20k data points for fine-tuning our stolen encoder. The performance of the stolen copy with only 10k queries (augmented to 20k data points) is similar to the stealing with 20k sentences.

**Vision:**
1. We propose a new method for the extraction of encoders which decreases the number of stealing queries. Therefore, we leverage semi-supervised learning based on MixMatch, which had previously only shown to be successful for stealing in supervised learning, and not for high-dimensional representations.
2. We propose a new defense that combines dataset inference with watermarking. We train the defended encoder on a public dataset and a small private subset, which is called a seed. The ownership is resolved by analyzing the likelihood of the private train vs test data points for a given encoder.

---

### Author Response · Authors · 2022-12-06
**Pending questions**

We would like to thank the reviewers for their feedback. The paper has definitely improved as a result. We would like to check one last time if there are any pending questions that we have not adequately addressed.

---

### Author Response · Authors · 2022-12-13
**Final feedback**

Dear Reviewers,

Please, check our responses and let us know if they addressed your concerns.

Thank you for your time, The authors

---

### Decision · Program_Chairs · 2023-01-20

**Decision:**

Reject

**Justification For Why Not Higher Score:**

Work is clearly very incremental.

**Justification For Why Not Lower Score:**

N/A

**Metareview: Summary, Strengths And Weaknesses:**

The paper performs an empirical study of model-stealing attacks on transformers trained in a self-supervised way on vision or text data. It also studies a defense against model stealing that inserts private training data as a watermark, and uses dataset inference to identify this watermark.

The paper does not present any major technical novelties as both the attack and defense studied are straightforward adaptations of existing work. The paper is also not the first to study model-stealing attacks against transformer models (see references provided by reviewer e9r9). In addition, the paper could do a better job in clarifying the threat model (for example, the paper assumes an attacker with access to the exact model architecture but this is not formalized in a threat model as is common for papers in the security space).

Overall, the paper appears too incremental to be of great interest to the ICLR audience: this exact learning setting has not been studied empirically before, but it is also not clear from the results in the paper that studying it leads to major new insights or observations.